# Surfaceome dynamics reveal proteostasis-independent reorganization of neuronal surface proteins during development and synaptic plasticity

Marc van Oostrum [1,2,3], Benjamin Campbell [1,4], Charlotte Seng[1,5], Maik Müller [2,3], Susanne tom Dieck[6], Jacqueline Hammer [2], Patrick G. A. Pedrioli[2,3], Csaba Földy[1,5], Shiva K. Tyagarajan [1,4] & Bernd Wollscheid [1,2,3 ✉]

Neurons are highly compartmentalized cells with tightly controlled subcellular protein organization. While brain transcriptome, connectome and global proteome maps are being generated, system-wide analysis of temporal protein dynamics at the subcellular level are currently lacking. Here, we perform a temporally-resolved surfaceome analysis of primary neuron cultures and reveal dynamic surface protein clusters that reflect the functional requirements during distinct stages of neuronal development. Direct comparison of surface and total protein pools during development and homeostatic synaptic scaling demonstrates system-wide proteostasis-independent remodeling of the neuronal surface, illustrating widespread regulation on the level of surface trafficking. Finally, quantitative analysis of the neuronal surface during chemical long-term potentiation (cLTP) reveals fast externalization of diverse classes of surface proteins beyond the AMPA receptor, providing avenues to investigate the requirement of exocytosis for LTP. Our resource (neurosurfaceome.ethz.ch) highlights the importance of subcellular resolution for systems-level understanding of cellular processes.

[1] Neuroscience Center Zurich, Zurich, Switzerland. [2] Institute of Translational Medicine (ITM), Department of Health Sciences and Technology, ETH Zurich, 8093 Zurich, Switzerland. [3] Swiss Institute of Bioinformatics (SIB), Lausanne, Switzerland. [4] Institute of Pharmacology and Toxicology, University of Zurich, Zurich, Switzerland. [5] Laboratory of Neural Connectivity, Faculties of Medicine and Natural Sciences, Brain Research Institute, University of Zurich, Zürich 8057, Switzerland. [6] Max Planck Institute for Brain Research, Frankfurt, Germany. ✉email: wbernd@ethz.ch

Assembly of neurons into higher order circuits via synaptic connectivity enables the mammalian brain to perform incredibly complex cognition. The quality, quantity, and interactions of cell-surface proteins drive neuronal development and activity-dependent synaptic plasticity[1–5]. Emerging evidence suggests that the diversity of extracellular proteins is crucial for synapse formation and wiring specificity[6]. Although the molecular mechanisms remain elusive, it has been hypothesized that different surface molecules act in a combinatorial fashion[7]. In order to understand, manipulate, and model the underlying molecular principles, the key cell-surface players and their dynamic interactions need to be identified and characterized[8].

Comprehensive analysis of the neuronal surface-exposed proteome, referred to as the surfaceome, remains challenging mainly for technical reasons. Global protein analyses have provided inventories of protein expression across mouse brain regions and cell types[9,10], characterized proteome dynamics during neuronal differentiation[11], and identified newly synthesized proteins during homeostatic scaling[12,13]. Proteomics analysis using mass spectrometry (MS) in combination with chemical labeling or fractionation schemes have been used to study the composition of synaptic vesicles[14], the postsynaptic compartment[15–17], the pre-synaptic active zone[18], synaptosomes[19], the synaptic cleft[20], the axonal proteome[21], interactomes of neuronal receptors[22–25], and the neuronal spatial proteome[26]. Similarly, the plasma membrane proteome of primary neuronal cultures has been investigated using metabolic labeling and subsequent enrichment of glycoproteins[27,28]. This approach enables quantitative comparison of the plasma membrane glycoproteome between different conditions, but cannot unambiguously define the acute surfaceome. The protein-level affinity enrichment employed precludes a priori separation of enriched surface proteins from nonspecific background, thus there must be a prior knowledge of surface localization to filter posthoc for known plasma membrane proteins. Furthermore, chemical labeling or fractionation strategies are labor intensive, especially for multiplexed applications with primary neuronal cultures. To quantitatively link spatial proteotypes to functional phenotypes, it is necessary to capture the dynamics of protein expression at the subcellular level[29].

Recently, we reported a miniaturization and automation of the Cell Surface Capture (autoCSC) method, enabling sensitive and multiplexed interrogation of the surfaceome landscape by direct identification of extracellular N-glycopeptides[30,31]. On living cells, cell-surface carbohydrates are tagged with cell-impermeable bio-cytin-hydrazide. After cell lysis and tryptic digestion, glycopeptides are enriched and released by peptide:N-glycosidase F (PNGase F) treatment. This leaves a deamidation within the N-X-S/T consensus sequence of formerly N-glycosylated peptides, enabling specific identification of extracellular N-glycosylation sites and quantification of protein abundance with subcellular resolution by MS[32]. The main advantage of this chemoproteomic strategy focusing on glycoproteins is the high surface specificity[33–35], ultimately providing snapshots of the acute surface-residing pool.

Here, we perform a quantitative analysis of the surfaceome during neuronal development and synapse formation using autoCSC. We describe time-resolved surface protein abundance profile clusters that match developmental stage-specific patterns. Integrative data analysis reveals system-wide protein surface translocation in response to plasticity paradigms and demonstrates that surfaceome protein abundance can correlate or uncouple from the total cellular proteostasis depending on the time resolution. The data indicates that localized proteotype maps provide functional insights into spatiotemporal controlled biological processes, such as synapse formation, which are typically masked in broad omics-style approaches.

## Results

**Quantification of synapse development in culture.** Primary neuronal cultures have been used to study the fundamental aspects of neuronal development, synapse formation, and synaptic plasticity. Upon culture in vitro, neuronal cells synchronize and differentiate through well-defined stages into a network containing functional synapses[11,33,34]. We first evaluated the time scale of synapse formation during neuronal development in cortical cultures (Supplementary Fig. 1). Between 10 and 18 days in vitro (DIV), excitatory and inhibitory synapse numbers increased more than 5-fold, followed by a slight reduction at 20 DIV (Supplementary Fig. 1). Throughout this period, on average 72% of synapses were excitatory (Supplementary Fig. 1). These data demonstrate that the majority of excitatory and inhibitory synapses form between 10 and 18 DIV; therefore, this is the relevant time window for synapse formation in culture.

To further characterize our cortical culture system, we established the relative abundances of neurons and astrocytes. Of all cells that could be categorized based on Gfap and Map2 immunofluorescence, we found an average of 81% to be neurons (Supplementary Fig. 1). Of those, we classified 23% on average as inhibitory neurons (Supplementary Fig. 1).

**Quantitative analysis of the neuronal surfaceome.** In order to quantify surfaceome dynamics during neuronal differentiation and synapse formation, we used autoCSC to take snapshots of the acute surface exposed N-glycoproteome every 2 days from 2 to 20 DIV (Fig. 1a). For each time point we prepared three biological replicates of cortical cultures. In order to preclude glial over-growth, the cultures were treated with cytosine arabinoside on DIV4. We normalized the total peptide amounts in all replicates prior to automated N-glycopeptide capture. The eluted, formerly glycosylated and extracellular peptides, were analyzed by DIA-MS and peptide-centric signal extraction[32,35,36]. Overall, we identified 1786 unique extracellular N-glycosylation sites (unique asparagines located within the N-X-S/T glycosylation motif of proteotypic peptides) and grouped them into 1024 protein groups for quantification (Supplementary Data 1). We compared the identified sites with the UniProt database and found that 441 sites had been annotated previously while 1345 sites were novel (Supplementary Fig. 2 and Supplementary Data 1). For the 1024 protein groups, we found numerous significantly enriched Gene Ontology (GO) annotations related to the cell surface and neuron function (Supplementary Fig. 3). For the majority of protein groups (76%) we obtained quantitative values for all time points. This overlap increased to 85% when the first two time points were excluded (Fig. 1b). These data provide a comprehensive picture of extracellular N-glycosylation sites and surfaceome proteins from primary cortical cultures. Furthermore, these results indicate that there are relatively few qualitative alterations to the surfaceome during maturation in culture (Fig. 1b), suggesting that regulation rather affects the quantitative surface abundance.

We first compared protein group quantities within replicates of the same time points and found high Pearson correlation coefficients with a median of 0.965 (standard deviation: 0.025) indicating high quantitative reproducibility (Supplementary Fig. 4). Next, we created a matrix of correlation coefficients comparing the protein group quantities at all time points (median among replicates) against each other (Fig. 1c and Supplementary Data 1). Between neighboring time points (e.g., two DIV vs. four DIV) we found generally high correlations (0.93–0.97). The lowest correlation coefficient of 0.73 was between the most distant time points (2 DIV vs. 20 DIV). Comparing all time points with 20 DIV, we observed that the correlation gradually increased with decreasing distance to 20 DIV, reaching the midpoint at six DIV.

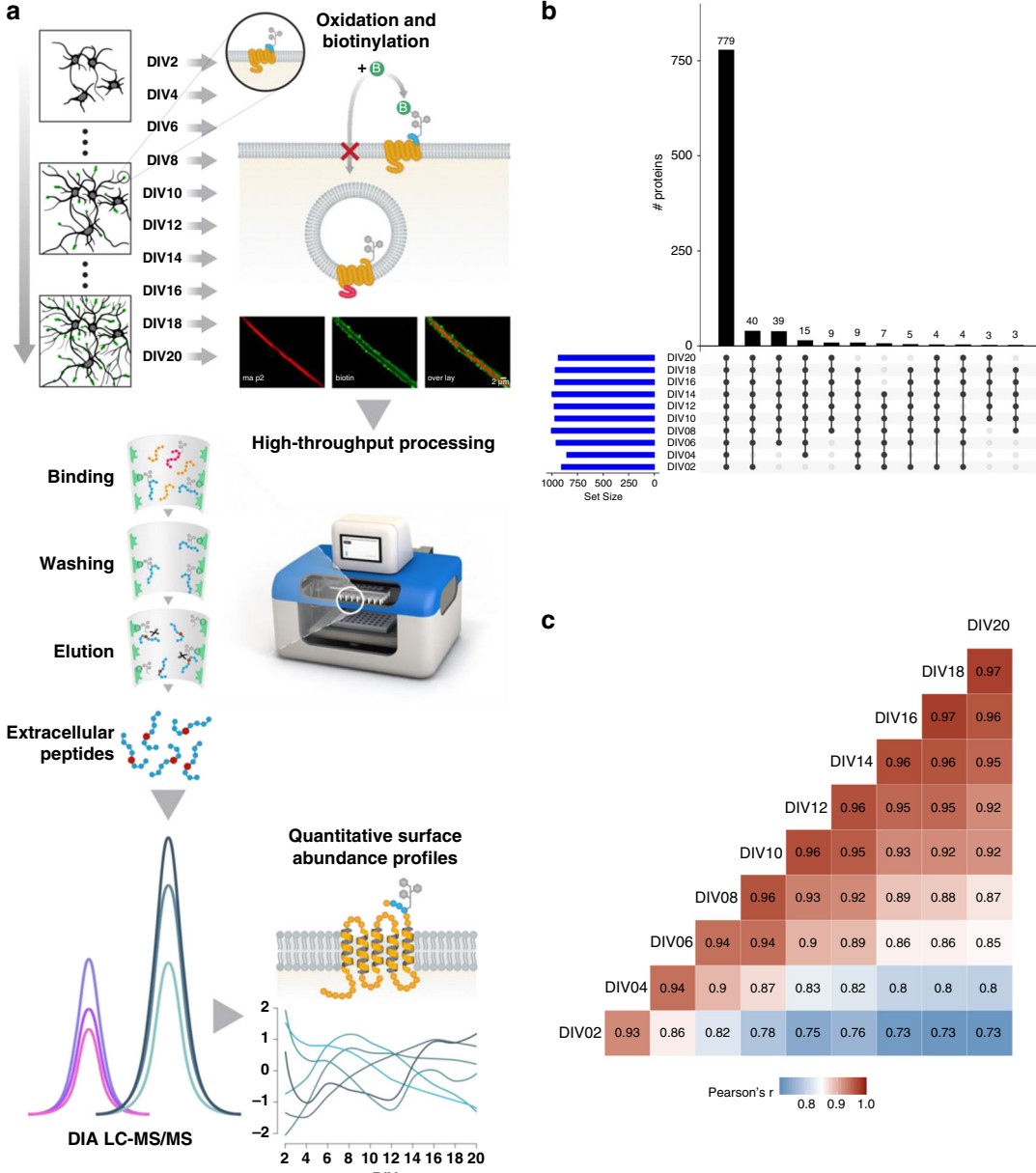

**Fig. 1 Quantitative analysis of the neuronal surface using autoCSC. a** Cultured cortical neurons were collected every second day from 2 to 20 DIV and subjected to autoCSC surface labeling. Live cells were oxidized under mild conditions with sodium-meta-periodate and subsequently labeled with cell-impermeable biocytin-hydrazide. After cell lysis and tryptic digestion, the resulting peptides were subjected to automated processing on a liquid handling robot. Repeated aspiration through filter tips containing streptavidin resin captures biotinylated *N*-glycopeptides. After extensive washing, peptides were released using PNGase F, which catalyzes the cleavage of asparagine-linked glycans. Cleavage leaves a modification (deamidation) at asparagines that can be identified using high-resolution MS. Initially labeled and extracellular peptides are identified by presence of deamidated asparagines within the NXS/T glycosylation consensus sequence, indicating both surface localization and glycosylation site. DIA and targeted feature extraction were used to quantify surface-protein abundances across multiple conditions, providing quantitative surface abundance profiles across neuronal development in culture. **b** Qualitative overview of the neuronal surfaceome composition illustrated by intersections of quantified proteins for each time point. **c** Correlation of all quantified protein abundance values per DIV (median per time point). Source data are provided as a Source Data file.

Furthermore, we found lower correlations (0.82–0.94) among the earlier time points (2–8 DIV) compared to time points after 14 DIV (0.95–0.97) (Fig. 1c). This indicates that the majority of quantitative changes occur during the early phases of development, notably before peak synaptogenesis.

Next, we identified proteins with significantly different surface abundance levels compared to the preceding time point. Across all comparisons we found 832 significant differences originating from 539 unique protein groups (Supplementary Data 1). The highest numbers of significant differences were found for the comparisons involving 4, 6, and 8 DIV (220, 190, and 132 significantly different proteins, respectively) (Supplementary Data 1 and Supplementary Fig. 5). This demonstrates that there is a substantial remodeling of the neuronal surface (>50% of all measured proteins) during differentiation in culture, with the most quantitative protein surface abundance changes occurring during the early stages. Notably, most surface protein abundance changes occurred prior to the time window for synapse formation, and few changes were observed after ten DIV.

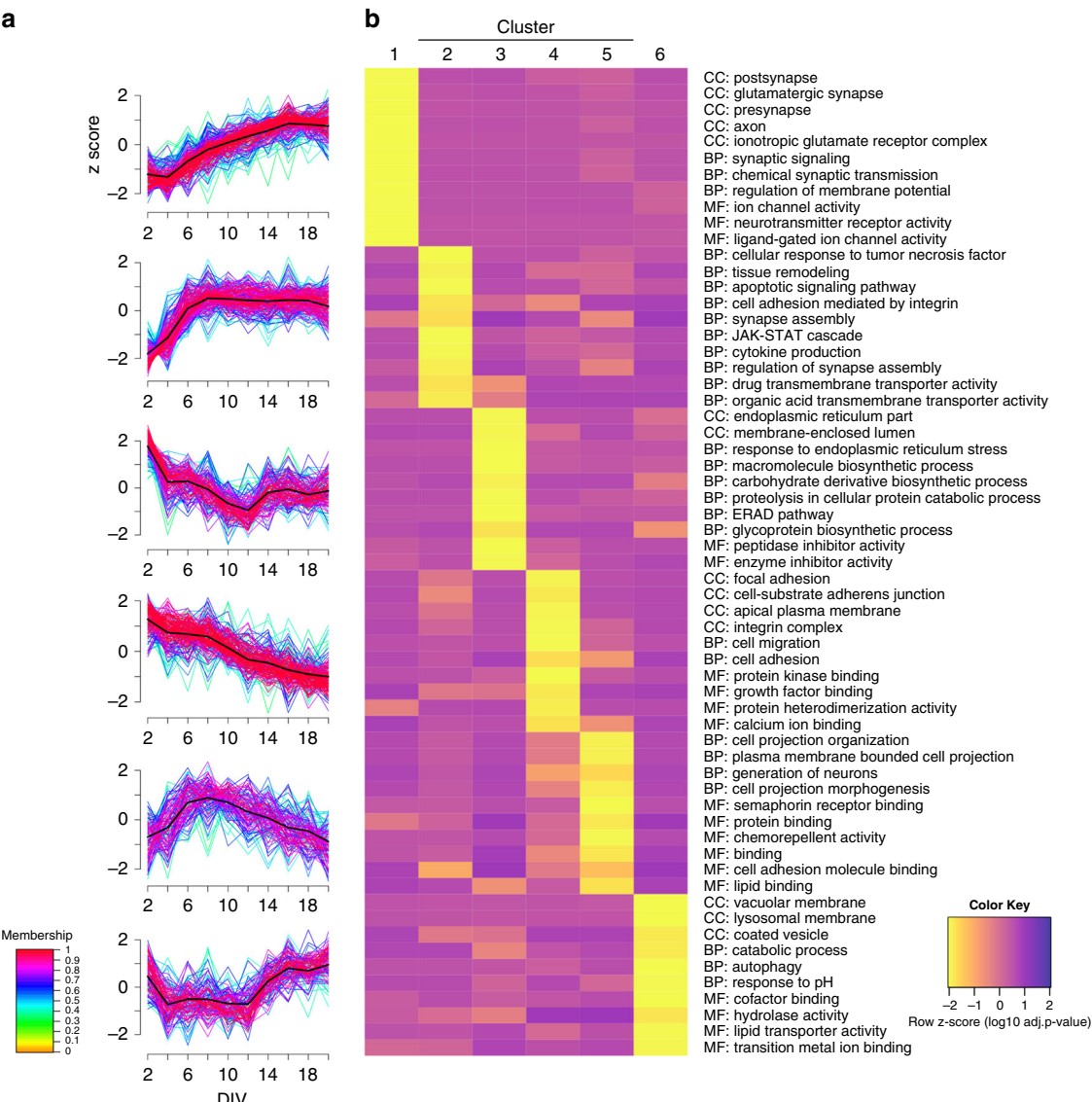

**Fig. 2 Surface abundance profiles reveal developmental stage specific patterns. a** Unsupervised c-means fuzzy clustering of surface abundance profiles reveals six distinct clusters. Colors indicate cluster membership. **b** Representative GO annotations for each cluster demonstrating that MF molecular functions, CC cellular components, and BP biological processes are enriched in at least one cluster ($p < 0.05$, Fisher's exact test). Source data are provided as a Source Data file.

In order to validate the results from autoCSC using an orthogonal methodology, we performed live-cell surface labeling for GABA and AMPA receptors at DIV6 and DIV16 and could confirm the significant increase in surface abundance with immunocytochemistry (Supplementary Fig. 6).

**Surface-abundance profiles reveal stage-specific patterns.** In order to investigate the diversity in surface abundance profiles over time, we used unsupervised fuzzy clustering of all proteins identified in at least seven time points[37]. We obtained six clusters with characteristic surface-abundance profiles (Fig. 2a, Supplementary Data 1, and Supplementary Fig. 7[38]). Clusters 1 and 4 include proteins that increased or decreased in abundances, respectively. Cluster 2 proteins increased in surface abundance across the time course but have an initially steeper abundance increase compared to proteins in cluster 1. Clusters 3, 5, and 6 featured dynamic profiles with alternating directionality.

To determine whether annotations of the protein group clusters reflect functional requirements of specific developmental stages, we retrieved ontology annotation data for the proteins of each cluster and selected significantly enriched annotations ($p < 0.05$) that showed specific enrichment in one cluster (Fig. 2b). The abundances of cluster 2 proteins steeply increased until eight DIV and remained relatively stable afterward. Interestingly, we found specific enrichment of the terms "synapse assembly" and "regulation of synapse assembly" for cluster 2, indicating that many synapse assembly factors reach the maximum surface abundance several days before the onset of synapse formation (Supplementary Fig. 1). Cluster 1 proteins increased in abundance starting at around four DIV and reached a plateau at around 16 DIV. Terms associated with synapse localization (e.g., "postsynapse", "presynapse", "glutamatergic synapse") and synaptic function (e.g., "synaptic signaling", "chemical synaptic transmission", "ion channel activity", "neurotransmitter receptor activity", "regulation of membrane potential") were specifically enriched in cluster 1 (Fig. 2b). This indicates that numerous synaptic proteins are increasingly trafficked to the cell surface days before the majority of synapse formation takes place.

Furthermore, we found that abundances of cluster 1 proteins reach the maximum surface intensity at 16 DIV, whereas synapse counts continued to increase until 18 DIV (Supplementary Fig. 1). Cluster 5 proteins also had increasing surface abundance at early time points, reaching a peak at 6–8 DIV, followed by a continuous decrease through 20 DIV (Fig. 2a). This raises the question whether proteins in this cluster have functions associated with the period at the cluster peak. DIV 5–8 is the period associated with outgrowth and branching of dendrites[21,33,34]. Indeed, we found the enrichment of proteins involved in cell projection (terms: "cell projection organization", "plasma membrane bounded cell projection", "cell projection morphogenesis") as well as "chemorepellent activity" and "semaphorin receptor binding" for proteins of cluster 5. Cluster 4 protein abundances continuously decrease over time and are associated with terms such as "cell migration", "growth factor binding", and "focal adhesion". Potentially these proteins are of importance during the initial 48 h in culture. Clusters 3 and 6 share a curious feature: both show an acute increase in abundance starting at 12 DIV (Fig. 2a), the time point when synapse formation increases considerably (Supplementary Fig. 1). Interestingly, both clusters are enriched in proteins associated with intracellular membranes, but the two clusters are enriched in membranes from different organelles (cluster 3: "endoplasmic reticulum part", "membrane enclosed lumen", cluster 6: "vacuolar membrane", "lysosomal membrane", and "coated vesicle"). These intracellular membranes potentially appear at the surface at increased rates due to alterations in surface trafficking. Associated annotations point to functions related to degradation or clearing of biomolecules, not only from the surface (cluster 3: "proteolysis in cellular catabolic process", "ERAD Pathway", cluster 6: "catabolic process", "autophagy", "hydrolase activity") but also the synthesis of new biomolecules (cluster 3: "glycoprotein biosynthetic process", "macromolecules biosynthetic process") (Fig. 2b). These findings could reflect acute changes in the dynamic interplay of biosynthesis, endosomal degradation, recycling, and surface trafficking, all important for control of synapse formation and modulation of synaptic plasticity[11,39–41]. Alternatively, some of these proteins may have been identified through labeling of small amounts of intracellular proteins from cell debris present in the culture supernatant, or very low levels of cell permeabilization by the linker. Altogether, these data illustrate distinct and dynamic surface-abundance profile clusters that reflect stage-specific functional requirements during neuronal development. Furthermore, the data indicate that numerous synapse assembly factors and synaptic proteins are produced and trafficked to the neuronal surface prior to the onset of synapse formation in culture.

**The in silico neuronal surfaceome interaction network**. The cluster analysis suggested that functionally related proteins have similar surface-abundance profiles during neuronal development in culture. We asked whether known protein complexes have higher correlations in surface-abundance profiles across our time series. We calculated Spearman correlation coefficients of the surface abundances for all pairwise combinations of surface proteins and found an approximately symmetric distribution with a median of 0.04, indicating no correlation on average. In contrast, proteins known to be in a complex together[42] had a median correlation coefficient of 0.55 (Supplementary Fig. 8).

In order to provide a bird's-eye perspective on surfaceome connectivity, we created a map of the surfaceome, associated interactions, and developmental surface profiles (Fig. 3a, b) using interaction data obtained from the STRING database. For a more in-depth analysis of specific proteins of particular interest, we first

mapped ligand-gated ion channels (Fig. 3c)[43], consisting of glutamate and GABA receptor complexes. More than half of ligand-gated ion channels were grouped into cluster 1; almost three-quarters of ligand-gated ion channels belong to either cluster 1 or cluster 2 (Fig. 3d). Receptor subunits matching outside of the majority clusters could indicate a modulatory function of the receptor or subunit exchange during neuronal development. For example, $GABA_A$ receptor complex subunits are spread over four different clusters, reflecting underlying molecular and functional heterogeneity[44]. Next, we mapped trans-synaptic cell adhesion molecules[43] to the surfaceome network (Supplementary Fig. 9). This analysis revealed several subnetworks of known protein families. For example, the ephrin family forms a subnetwork with many members associated with clusters 4 and 5. Similarly, cadherins are highly interconnected and mostly associated with clusters 1 and 2.

Finally, we examined proteins encoded by mRNAs that are enriched in neurons compared to astrocytes or vice versa. From cell-type resolved RNA-seq data[45], we selected protein groups matching transcripts that are enriched 5-fold or more in one cell type compared to the other and mapped them to the STRING network. Comparing the relative sizes of clusters, we observe that they are differentially populated for highly cell type specific transcripts (Supplementary Fig. 9). Interestingly, some subnetworks feature surface proteins from both cell types, for example the ephrins and semaphorins, potentially hinting at inter-cell type molecular interactions. In summary, this map of the neuronal surfaceome combines connectivity and developmental dynamics as a resource for the neuroscience community.

**Differential regulation of surface and total cellular pools**. A dynamic interplay of biosynthesis, surface trafficking, and endosomal degradation and recycling provides the cell surface with the necessary quantity of surface proteins required for neuronal development and synaptic plasticity[41]. In order to investigate this dynamic process, we subdivided cell surface proteins into two theoretical pools: a total pool composed of proteins present at the surface and intracellularly, and the surface pool as measured by autoCSC. Comparison of these pools would enable investigation of different levels of organizational principles: for example, whether observed changes in surface abundance are explained by changes in the total protein abundance or by altered surface trafficking. In order to obtain data on the total pool, we performed proteotype analysis on the same samples described above by measuring the non-$N$-glycosylated tryptic peptides by DIA-MS (Fig. 4a)[32,35,36].

In total, we quantified approximately 7400 protein groups and identified significantly differentially expressed proteins using pairwise testing against the preceding time point as in autoCSC (Supplementary Data 2). Across all comparisons we found 3061 significant differences matching 2193 unique protein groups (Supplementary Fig. 5). Thus, about 30% of the measured proteome is differentially expressed across the time course of neuronal differentiation (Supplementary Data 2). By comparison to the autoCSC data, we found that the neuronal surfaceome is more dynamic than the total proteome. About 53% of surfaceome proteins were differentially regulated, potentially due to additional regulation on the level of surface trafficking.

For a more direct comparison, we matched protein groups from the surface and total pool; information on both pools was available for about 650 protein groups (Supplementary Data 2). First, we asked whether the protein abundance profiles across the time course of neuronal development correlate between surface and total pools. We calculated Spearman correlation coefficients for each protein and found that 78% had a positive value with a

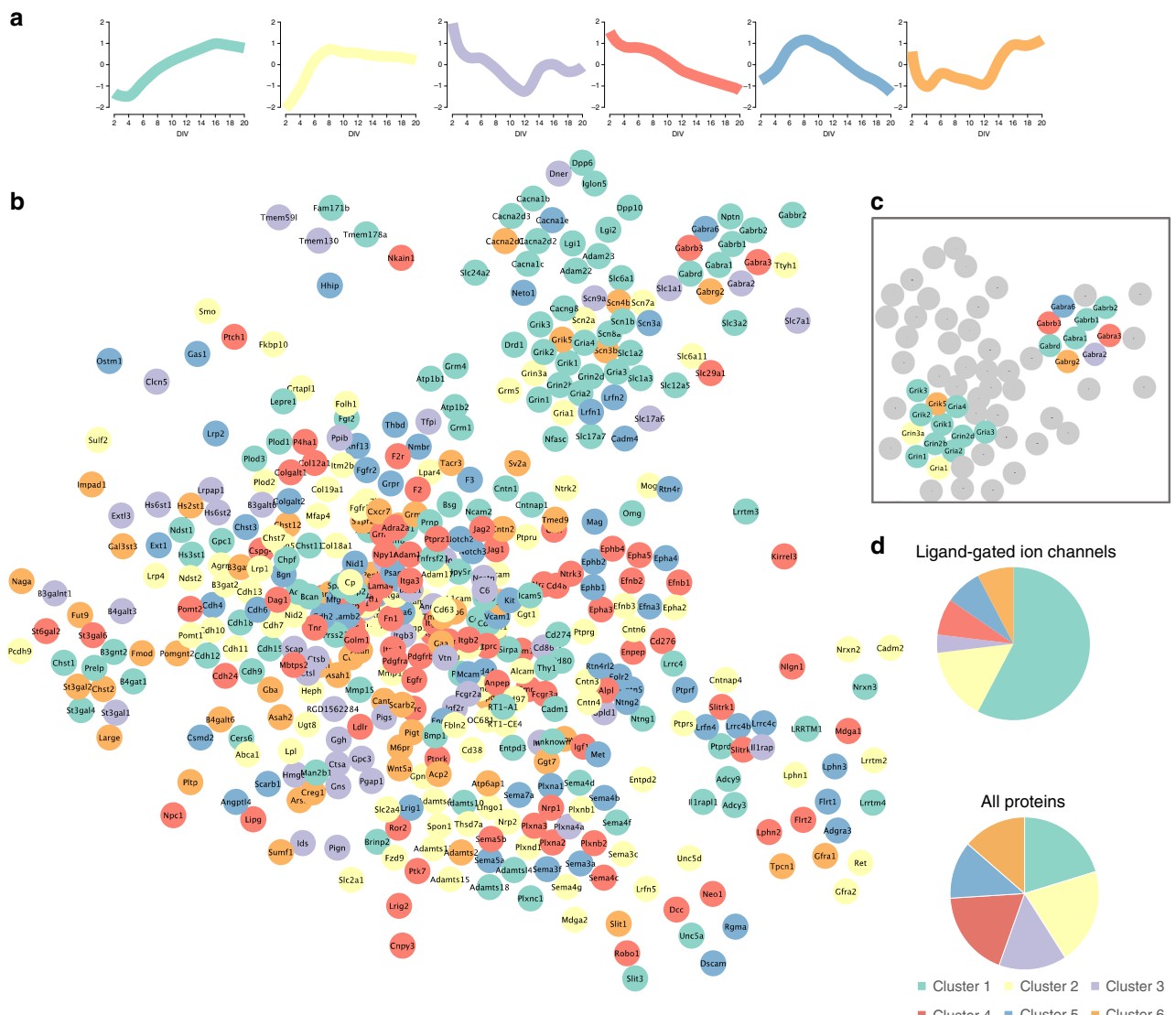

**Fig. 3 The neuronal surfaceome interaction network. a** Cluster legend showing median cluster profile with the color code used in the string network **b** String network representation of the largest connected cluster of identified surface proteins; edges represent string confidence of 0.7[93]. **c** Network representation of ligand-gated ion channels is shown on the top right. **d** Pie charts of cluster contributions to ligand-gated ion channels and all surfaceome proteins. Source data are provided as a Source Data file.

median correlation coefficient of 0.48, indicating a strong positive correlation for the majority of surface proteins across the neurodevelopmental time course (Fig. 4b and Supplementary Data 2). Thus, over the entire 20 days, for many proteins the total abundance profile reflects the surface abundance.

The relevant time scale for modulation on the level of surface trafficking might be shorter than the 18-day interval evaluated. In order to investigate the relationship of surface and total abundances over time, we calculated pairwise comparisons between time points (Supplementary Data 2). First, we compared the abundance changes between the two most distant time points, 2 and 20 DIV, and visualized them in a scatterplot (Supplementary Fig. 10). The majority (82%) of all significantly regulated proteins have agreeing directionality (i.e., they were found in quadrants 1 or 3) (Supplementary Fig. 10). Furthermore, we found a 43% overlap of significantly different proteins and a correlation coefficient of 0.73 for the comparison between the most distant time points (Fig. 4c, d). Next, we asked how these parameters change when the time window of observation was decreased. Interestingly, the median correlation coefficient

decreased with decreasing time between measurements, to a median of 0.26 for neighboring time points (Fig. 4c). Similarly, the overlap of significantly different proteins between surface and total pools decreased from 43 to 13% with decreasing time between compared measurements (Fig. 4d). Together, this demonstrates that for many proteins the total abundance pool correlates with the surface pool during neuronal differentiation in culture, and changes in surface trafficking influence the surface abundance generally on shorter time scales.

**Implications of uncoupling of surface and total abundance.** If surface and total abundance of a protein are highly correlated, there is likely little regulation of surface trafficking. Conversely, if surface and total abundance are uncoupled, there is presumably systematic modulation of surface abundance by differential trafficking. Therefore, we sought to systematically identify the proteins with the most deviation between surface and total abundance during neurodevelopment in culture. Proteins in the previously described protein clusters (Fig. 2a) generally had temporal responses that fit an impulse model; this is a parametric

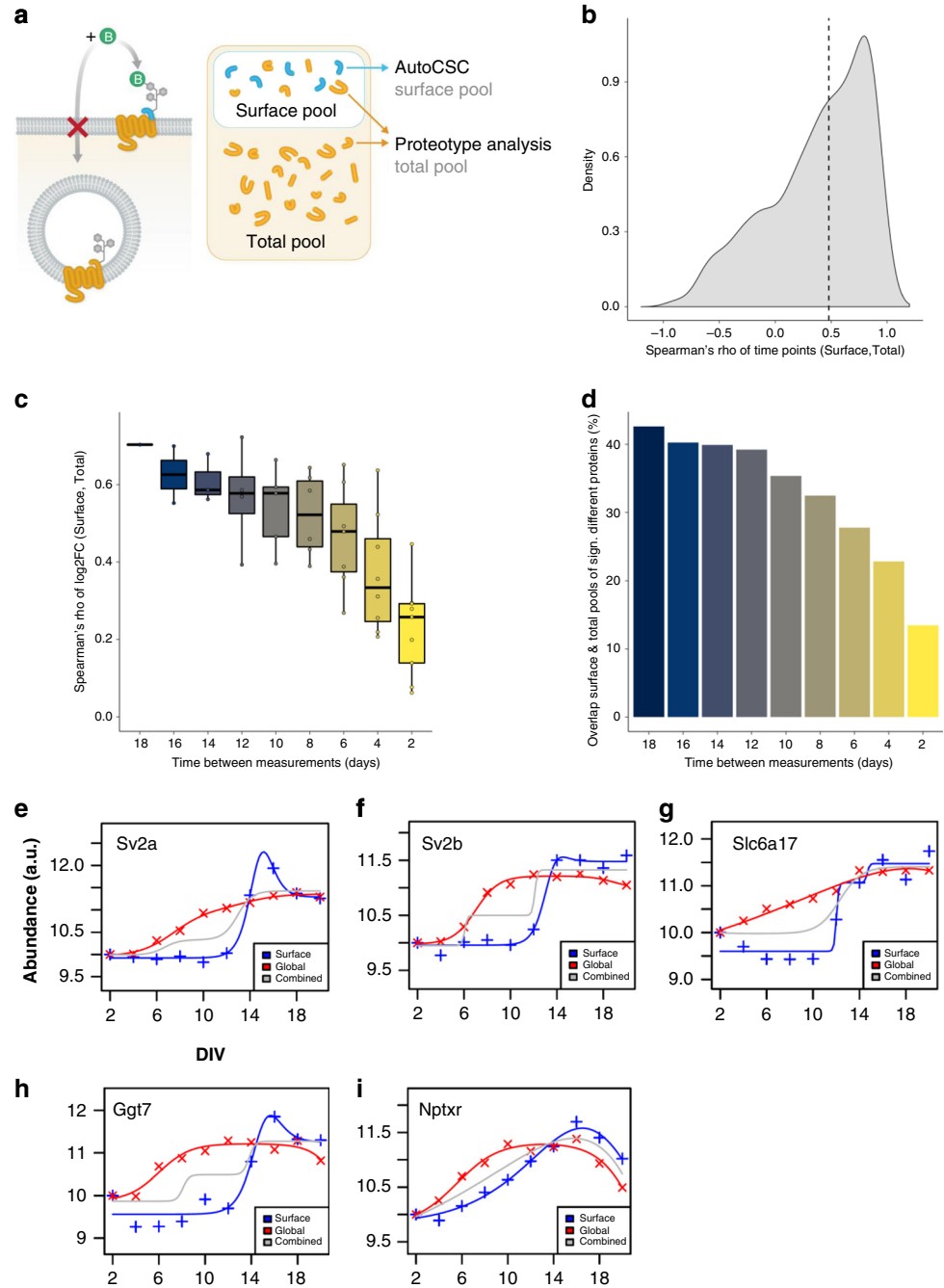

**Fig. 4 Correlation of surface with total abundance during differentiation. a** Comparison of the surface pool against the total protein pool. Peptides analyzed by autoCSC (blue) are surface exposed and thus specifically report on the acute surface pool. Tryptic peptides from the same samples (orange) report the total protein abundance including both intracellular and surface-localized proteins. **b** Correlation of surface and total abundance protein pools. Spearman's rho of surface and total abundance time series, median indicated with dashed line. **c** Boxplots of distribution of Spearman's rho for log2 fold-changes between indicated time intervals for surface and total protein. Boxes indicate median and percentiles (25th and 75th). **d** Bar chart of counts of overlapping significantly different proteins (fold-change > 1.5 and $p < 0.05$) for comparisons between surface and total pools with respect to interval times. **e–i** Impulse model fit for data for **e** Sv2a, **f** Sv2b, **g** Slc6a17, **h** GGt7, and **i** Nptxr. Crosses indicate the normalized median abundance values of three replicates for each time point for surface (blue) and total (red) pools. Lines indicate impulse model fits; gray lines are fits to combined data from surface and total pool analyses. Source data are provided as a Source Data file.

model with an early response followed by a second transition to a steady state[46]. Using impulse models[47], we observed very good agreement between modeled surface and total abundance profiles for many proteins (Supplementary Fig. 11). This suggests that for these proteins changes in surface abundance are predominantly determined by the total abundance, without detectable regulation on the level of surface trafficking. For 128 proteins we observed

differential regulation (adj. $p$-value < 0.01, Supplementary Data 2). Interestingly, synaptic vesicle proteins SV2a and SV2b were both among the significantly regulated proteins. Synaptic vesicles are assembled intracellularly and only become surface exposed, once exocytosis is increased during initialization of synaptic activity. There was a continuous increase in total abundance of SV2a and SV2b beginning early in development.

Amounts in the surface pools, however, did not change until the onset of synapse formation (12 DIV), and then we observed significant increases in surface abundance that reached steady state at the late time points (Fig. 4e, f). For three other proteins, Slc6a17, Ggt7, and the neuronal pentraxin receptor Nptxr[48] we found similar profiles, indicative of intracellular pools that become increasingly surface exposed upon synapse formation (Fig. 4g–i). In summary, these data show that differences between regulation of surface and total abundance can reflect protein function and subcellular organization.

**Homeostatic plasticity induces remodeling of the surfaceome.** Changes in synaptic strength are referred to as synaptic plasticity and thought to underlie learning and memory formation. Long-term potentiation (LTP) acts on individual synapses and is characterized by a long-lasting increase in synaptic strength upon high-frequency stimulation[2,3,49]. In contrast, homeostatic plasticity, or synaptic scaling, can affect the strengths of multiple synapses proportionally, potentially promoting network stability[50]. We asked how the quantitative composition of the neuronal surfaceome adapts in response to LTP and synaptic scaling.

To investigate homeostatic plasticity, we elicited synaptic upscaling and downscaling in mature cortical neuronal cultures (DIV 21) by treatment with the neurotoxin tetrodotoxin (TTX, 1 μM) or with the GABA$_A$ receptor antagonist bicuculline (BIC, 40 μM), respectively, for 24 h[12,51]. Increased miniature excitatory postsynaptic currents (mEPSCs) upon TTX and decreased mEPSCs upon BIC treatment were previously reported for 24 h treatments[12]. In line with these results, we found increased fast EPSC amplitudes (Fig. 5a, b) as well as synaptic burst frequencies (Fig. 5c) comparing TTX with BIC treatment, indicative of differential synaptic scaling.

As surface SV2a and SV2b levels increased with synapse formation and neuronal activity, we expected to observe a decrease in surface abundance upon TTX treatment. Indeed, using autoCSC, we found a significant reduction of both SV2a and SV2b compared with untreated controls (Supplementary Fig. 12), indicating a net decrease of neuronal synaptic vesicle release. In contrast, upon downscaling of GABAergic neurotransmission using BIC, which leads to onset of rapid neuronal firing, there was a significant increase in surface SV2a and SV2b (Supplementary Fig. 12), suggesting neuronal firing events lead to a net increase in synaptic vesicle fusion events. Importantly, autoCSC is expected to report on the status with TTX or BIC present, while our electrophysiological recordings were performed after the 24 h treatment in absence of TTX or BIC. Considering the previous findings that homeostatic synaptic scaling predominantly influences the postsynapse[52], the observed changes in SV2a/b likely are a direct consequence of TTX or BIC treatment, rather than homeostatic scaling.

During synaptic scaling, as during neuronal development, the qualitative surfaceome composition did not significantly change. Overall, we quantified 864 protein groups, with 850 identified in all conditions. In a quantitative analysis, 246 proteins were significantly different compared to an untreated control (Fig. 5d and Supplementary Data 3), indicating extensive quantitative remodeling of the neuronal surface. Of these, the levels of 102 proteins were changed in both upscaling and downscaling with coordinate directionality. This indicates a role in a general response to global changes in neuronal activity, regardless of the sign of plasticity (Fig. 5d). Functional annotation analysis of all regulated proteins revealed significant enrichment in terms related to synaptic function (e.g., "sodium channel activity", "transmembrane receptor kinase activity", "membrane depolarization", and "action potential"). Interestingly, the vast majority

of proteins associated with an enriched term had positive fold-change and agreement in directionality between opposite scaling paradigms (Supplementary Fig. 13).

Previous studies have mechanistically linked calcium influx with synaptic upscaling[53,54]. In support of this, we found that surface abundances of a number of voltage-gated calcium channels were significantly modulated in response to both homeostatic plasticity paradigms (e.g., Cacna2d3, Cacna1d) or only to downscaling (e.g., Cacna1c) (Fig. 5d). EphA4 has been shown to mediate synaptic downscaling[55], and we observed a significant increase in levels of this protein upon both upscaling and downscaling (Fig. 5d). To specifically investigate proteins with bidirectional regulation comparing TTX with BIC treatment, we performed significance testing and identified 68 proteins with differential abundance (Fig. 5e and Supplementary Data 3). Most of these proteins either increased in surface abundance during upscaling and decreased in surface abundance after downscaling or vice versa, indicating that the change was opposite the polarity of stimulation. Interestingly, both receptor-type tyrosine-protein phosphatase-like proteins Ptprn and Ptprn2 changed in surface abundance upon synaptic scaling (Fig. 5e), suggesting a role in this process. Although Ptprn does not change upon upscaling, it significantly increases in the surface abundance upon downscaling. The opposite was true for Ptprn2, as we observed no change during downscaling but a decreased surface abundance in response to TTX (Fig. 5e).

Similarly, we observed remarkable and unique differential regulation for the three neuronal pentraxins Nptx1, Nptx2, and Nptxr (Fig. 5e). Nptxr was less abundant on the surface after upscaling and more abundant after downscaling. Upscaling elicited no change in Nptx2 surface abundance, but downscaling led to a significant increase in surface abundance. Nptx1 increased in surface abundance upon both upscaling and downscaling (Fig. 5e). In agreement with our data, it was previously reported that surface abundance of Nptx1 increases after 24 h of TTX treatment and that inhibition of Nptx1 expression blocks synaptic upscaling[54].

We did not detect significant changes in any condition for any subunit of the AMPA receptor (Fig. 5e). However, three metabotropic glutamate receptors had significantly polarized responses. Grm1, Grm4, and Grm5 had different responses to upscaling than to and downscaling, predominantly due to lower surface abundance upon BIC treatment (Fig. 5e). Finally, we identified two related proteins, Sidt1 and Sidt2, with strong and different responses to synaptic scaling. Sidt1 abundance increased upon upscaling but decreased after BIC treatment. In contrast, Sidt2 abundance decreased in both conditions, albeit significantly more strongly during downscaling (Fig. 5e). These findings suggest a connection between synaptic plasticity and cellular transfer of RNA as Sidt1 and Sidt2 bind and transport double-stranded RNA[56,57].

Next, we asked whether the changes we observed on the neuronal surface were reflected in the total proteotype. Across all comparisons, 28% of all quantified surface proteins changed in abundance, compared to 8% for the total proteome (Supplementary Data 3). Only 2% of the changes in surface abundance were accompanied by a significant change of the same proteins in total abundance after 24 h (Fig. 5f and Supplementary Data 3). We also noted that the surface proteins that changed in abundance, the fold-change difference was higher for the surface abundance change than for the total abundance change (Supplementary Fig. 14). For the few surface proteins that increased both on surface and total abundance, we found agreement with previous reports. During synaptic upscaling total and surface abundance levels of Nptx1 increased (Fig. 5e), which recapitulates previous findings achieved with orthogonal approaches[54]. For synaptic

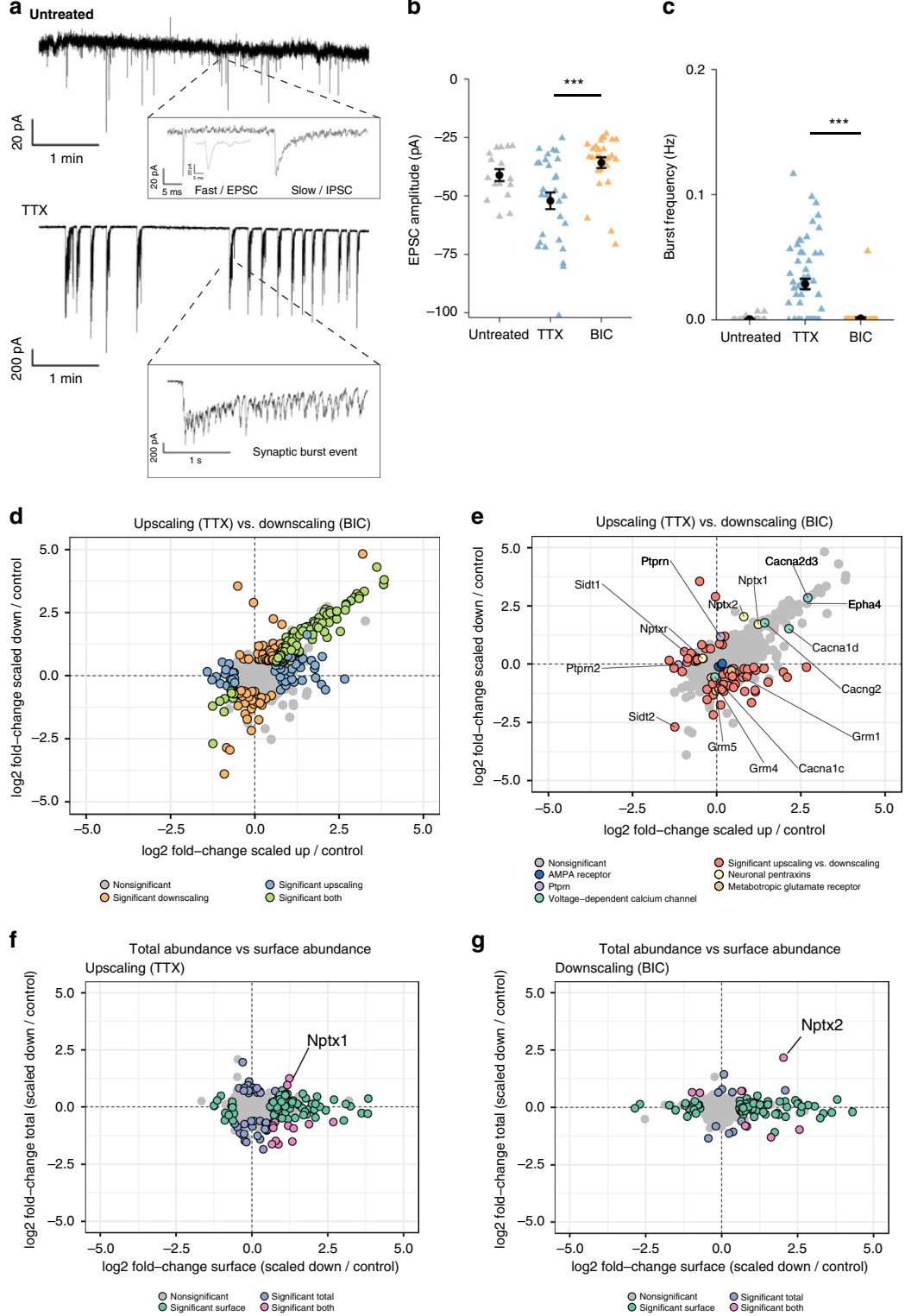

downscaling, our data on Nptx2 (Fig. 5f), provides supporting an evidence for polarized roles of neuronal pentraxins in homeostatic plasticity.

In order to validate the results from autoCSC using an orthogonal methodology, we performed live-cell surface labeling followed by immunocytochemistry for GABA_A receptor subunits gamma-2 and alpha-1. For the gamma-2 subunit, we could confirm the significant increase in surface abundance upon both synaptic upscaling and downscaling. For the GABA_A receptor subunit alpha-1, we found no significant difference by immunocytochemistry, in line with our results from autoCSC (Supplementary Fig. 15).

Together, these results provide a comprehensive picture of dynamic surfaceome changes in response to homeostatic plasticity, recapitulating previous knowledge, and revealing new opportunities for in-depth functional investigation. Importantly,

**Fig. 5 The neuronal surface is altered during homeostatic plasticity. a** Upper panel: Example electrophysiological recording from an untreated cell. Insert shows magnification of neighboring fast (EPSC) and slow (IPSC) synaptic events. The fast EPSC is further magnified for clarity. Lower panel: Example recording trace from a TTX treated cell. Insert shows magnification of a synaptic burst event. **b** Plot shows averaged fast EPSC amplitudes for each cell (triangles) and their averages (mean ± s.e.m) for untreated ($n = 16$), BIC ($n = 27$), and TTX ($n = 29$) treated cells. Untreated vs. BIC $p = 0.086$, Untreated vs. TTX, $p = 0.043$. Mann–Whitney test. ***$p < 0.001$ **c** Plot shows the frequency of synaptic bursts for each cell (triangles) and their averages (mean ± s.e.m) for untreated ($n = 38$), BIC ($n = 55$), and TTX ($n = 56$) treated cells. Untreated vs. BIC, $p = 0.17$, untreated vs. TTX, $p < 0.001$. Mann–Whitney test. ***$p < 0.001$ **d** Scatterplot comparing surface abundance changes during synaptic upscaling and downscaling. Significantly regulated proteins (fold-change > 1.5 and $p < 0.05$) are indicated by color. **e** Scatterplot comparing surface abundance changes during synaptic upscaling and downscaling as shown in **d** with protein groups of interest indicated. **f** Scatterplot showing abundance change (scaling vs. control) of synaptic upscaling comparing surface ($x$-axis) and total ($y$-axis) pools. Significantly regulated proteins (fold-change > 1.5 and $p < 0.05$) are indicated by color. **g** Scatterplot showing abundance change (scaling vs. control) of synaptic downscaling comparing surface ($x$-axis) and total ($y$-axis) pools. Significantly regulated proteins (fold-change > 1.5 and $p < 0.05$) are indicated by color. Source data are provided as a Source Data file.

the data indicate that there is dynamic reorganization of the neuronal surface during synaptic scaling that is largely independent of global protein abundance changes.

**Surface trafficking of diverse cargo during cLTP.** NMDA receptor-dependent long-term potentiation (LTP) is a well-studied form of synaptic plasticity, and there is a mechanistic understanding of early molecular events, prominently involving the modulation of synaptic AMPA receptor synaptic abundance and post-translational modifications[3,49]. One hallmark of LTP is surface insertion of AMPA receptors from intracellular pools, and an intact exocytosis machinery is crucial for LTP[58]. A three-step model for AMPA receptor synaptic retention has been proposed that involves exocytosis at extra/perisynaptic sites, lateral diffusion to synapses, and a subsequent rate-limiting diffusional trapping step[59]. While evidence for the importance of surface diffusion is accumulating[60–62], the role of AMPA receptors for the dependence of LTP on exocytosis has not been fully clarified, and there is speculation that currently unrecognized cargo could mediate the requirement for exocytosis[63–67]. While neuron cultures do not undergo LTP, they are frequently stimulated with cLTP treatments which reproduce hallmarks of LTP, such as AMPA receptor exocytosis, synaptic trapping of receptors and increase in synaptic transmission. To ascertain that our cLTP protocol leads to an increase in synaptic transmission, we performed electrophysiological recordings and confirmed increased fast EPSC amplitudes upon cLTP treatment (Fig. 6a, b). For a discovery-driven and comprehensive analysis of surfaceome changes, we treated cortical cultures with cLTP at DIV 14 followed by autoCSC surfaceome analysis 20 min later (Fig. 6a). As expected, we found that Gria1 and Gria2 had significantly increased surface abundance upon stimulation (Fig. 6c, d) indicating that the cLTP protocol triggers AMPA receptor exocytosis. Gria3 abundance was increased on the neuronal surface as well; although the effect reached statistical significance (Fig. 6e), it was slightly above the threshold after FDR-adjustment of $p$-values (Fig. 6e and Supplementary Data 4). Interestingly, there was a strong increase in surface abundance for AMPA auxiliary subunit stargazin (Cacng2) (Fig. 6f); this subunit promotes delivery of the AMPA receptor to the surface. Furthermore, stargazin is known for its ability to trap AMPA receptors at synaptic sites by direct interaction with PSD95[59,68]. In addition to AMPA receptor subunits, we identified 36 surface proteins that were significantly increased in abundance and four that were reduced in surface abundance after cLTP (Fig. 6g). Three enriched proteins had been previously connected to LTP by studies investigating their loss-of-function: Wnt-5a, the major prion protein Prnp, and the adenylate cyclase Adcy3. Inhibition of expression of any of these proteins leads to functional LTP defects[69–71]. The strongest responder was the receptor-type tyrosine-protein phosphatase

Ptprn, which is removed from the surface in response to cLTP (Fig. 6g). Interestingly, we found that Ptprn levels were increased on the surface during synaptic downscaling. These results suggest that Ptprn has a role in negative regulation of synaptic strength. In contrast, we observed increased surface abundance for another receptor-type tyrosine-protein phosphatase, Ptprc.

We grouped the proteins with significantly different surface abundances after cLTP treatment and visualized their interaction network (Fig. 6h). In addition to the AMPA receptor subunits and stargazin, three ion channels and ten other receptors increased in surface abundance upon cLTP. We identified receptors with known localizations from both presynaptic and postsynaptic sites (e.g., adenosine receptor A1 and cannabinoid receptor 1) and of various neurotransmitter types. The majority of receptors identified were previously described as postsynaptic G-protein coupled receptors (i.e., adhesion G protein-coupled receptor B1, D(1A) dopamine receptor, 5-hydroxytryptamine receptor 2C, muscarinic acetylcholine receptor M4, and alpha-2A adrenergic receptor) and interact with second messenger Adcy3, which was also increased in abundance. In addition to receptors, ten secreted proteins significantly changed in abundance, including semaphorin-3A, bone morphogenetic protein 3, and IgLON family member 5. Furthermore, four adhesion molecules, two solute carriers, two tetraspanins, and seven proteins with miscellaneous functions had altered abundance upon cLTP (Fig. 6h). Some of the identified proteins suggest links between LTP and cellular processes that had not previously been associated with LTP. For example, Tmem110, which was trafficked to the surface upon cLTP, and was recently shown to regulate store-operated calcium entry at junctions that spatially connect the plasma membrane with the endoplasmic reticulum[72]. Furthermore, the increased abundance of volume-regulated anion channel subunit Lrrc8b, which plays a central role in the maintenance of cell volume, potentially provides a link between early LTP and structural plasticity of dendritic spines[73,74]. The narrow time window for cLTP makes it unlikely that any of the observed increases on the surface are due to changes in total abundance of these proteins; however, specific degradation of surface proteins might occur concurrently with endocytosis. Therefore, we analyzed the total proteotype as before and found no proteins with significantly increased abundance (Supplementary Data 4). We did, however, identify proteins with reduced total abundance after cLTP, potentially due to rapid degradation. These proteins are mostly cytosolic, and we found no overlap with the identified surface proteins that decreased in surface abundance after cLTP (Supplementary Fig. 16).

We validated the results from autoCSC using an orthogonal methodology, namely neuromorphology. We selected five proteins for live-cell surface labeling and confirmed the increased surface abundance for the AMPA receptor, Bai1, and Adcy3 after cLTP (Fig. 6i, j). For Adcy3 we additionally observed localization

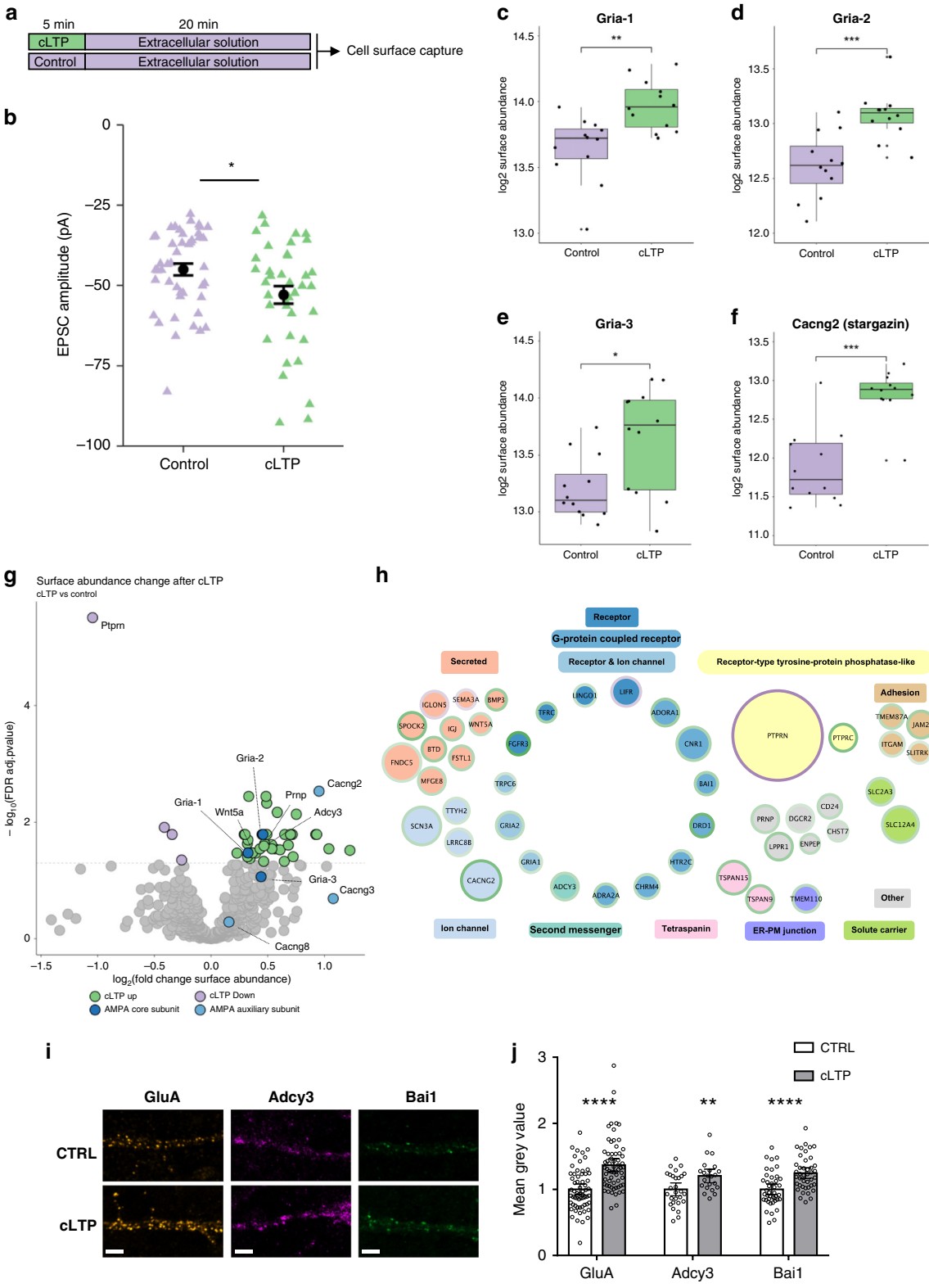

to synaptic sites, identified by puncta overlapping with pre-synaptic marker synapsin 1/2 (Supplementary Fig. 17). Additionally, we could confirm for two subunits of the GABA$_A$ receptor that they do not change significantly upon cLTP (Supplementary Fig. 17).

These data illustrate the diverse adaptations on the neuronal surface that occur in addition to alterations in AMPA receptor trafficking during cLTP. The identified molecules provide a number of avenues to investigate the molecular mechanisms underlying the dependency of LTP on exocytosis.

## Discussion

A comprehensive overview of how molecular dynamics at synapses enable construction and plasticity of neuronal circuits

**Fig. 6 Surface trafficking of diverse cargo during cLTP. a** cLTP paradigm **b** The plot shows averaged fast EPSC amplitudes for each cell (triangles) and their averages (mean ± s.e.m) for control (untreated, $n = 45$) and cLTP (glycine treated, $n = 38$) cells. Mann–Whitney test. **c**–**f** Boxplots of distributions of log2 surface abundances for replicates of cLTP ($n = 12$) and control ($n = 12$). Boxes indicate median and percentiles (25th and 75th). **g** Volcano plot of statistical significance (y-axis) and surface abundance change (x-axis). Horizontal line is located at an adjusted p-value of 0.05. **h** Proteins with significantly different surface abundances (fold-change > 1.5 and $p < 0.05$) after cLTP grouped into categories. Shape size indicates scaled $-\log10$ adjusted p-value. Border color indicates fold-change directionality (green > 0, purple < 0). Edges represent string confidence > 0.7. **i** Representative images of analysed primary dendrites before and after cLTP induction. Bars 5 µm. **j** Quantification of mean fluorescent intensity of antibody signal on the primary dendrite surface using antibodies against GluA: $n = 63$, Bai1: $n = 42$, Adcy3: $n = 28$ (CTRL) $n = 22$ (cLTP) cells/group. Means + 95% CI presented. Two-way T-test (Adcy3 + Bai1) or Mann–Whitney test (GluA). *$p < 0.05$, **$p < 0.01$, ****$p < 0.0001$. Source data are provided as a Source Data file.

has been elusive. The vast number of proteoforms that contribute to different processes in parallel, with generally unclear inter-dependencies, gives rise to a complexity that has been beyond comprehension[6,7,49,63,75]. Furthermore, cells are capable of ela-borate computations, even for reductionist model systems and isolated molecular events[8,76], raising the possibility that bioin-formatic models and analyses will be required to decipher non-trivial relationships. To implement such modeling, detailed knowledge about all molecules involved, their mechanisms of action and their interplay with appropriate spatial and temporal resolution is required. Targeted genetic manipulation and/or molecular biology combined with phenotypic measurements of synaptic structure and properties can provide in-depth informa-tion regarding the function, localization, and post-translational modifications of individual molecules, but this requires prior knowledge and a focus on a few genes of interest. Discovery-driven and system-wide approaches can inform on many mole-cules, but the measurement of global RNA or even protein abundance levels are an abstraction of the actual process of interest: the activity of a particular protein in executing a parti-cular spatiotemporally designated function. While there is no suitable all-in-one solution available, integration of different types of data will be required. Here, we used a time-resolved spatial chemoproteomics approach to evaluate neuronal development and synaptic plasticity. We used this approach to (i) map proteins at a defined subcellular structure, the surface, in a discovery-driven fashion, (ii) to perform system-wide quantification of both surface and total cellular abundance, and (iii) to quantify sub-cellular re-localization events.

We present an initial blueprint of how many, and which, proteins are present on the surface of neurons (see supplementary discussion). We observed very few qualitative differences on the surface during neuronal development in culture, which is rather surprising considering the profound morphological changes that cells undergo during this process[33]. In contrast, the quantitative analysis of neuronal surfaceome dynamics revealed substantial remodeling of the surface during development. Most of the dif-ferences were observed within the first week, which is the time period of the most drastic morphological changes[33]. We used unbiased clustering of protein abundance profiles to reveal developmental-stage-specific patterns. These data indicate that the acute functional requirements imposed on the neuronal surface are reflected in the quantitative surfaceome composition. Comparing the surface proteotype data with synapse counts along the developmental time series, we showed that the cluster enri-ched for synapse assembly proteins had a maximum surface abundance at eight DIV, four days before the increase of synapse counts. Furthermore, proteins in the cluster enriched with synaptic proteins peak in abundance 2 days before the maximum of synapse counts at 18 DIV. In a total proteotype analysis, Frese et al. found identical early dynamics of synaptic proteins[11]. Similarly, in vivo single-cell transcriptomic analysis of newborn cortical neurons revealed that 12-h old neurons express tran-scripts of synaptic genes[77]. Together, this suggests that many

synaptic proteins are produced and trafficked to the surface before synapses are formed and only later are these proteins organized into synaptic microdomains by surface diffusion. The existence of such a mechanism has been proposed based upon the finding that basic circuit connectivity, including dendrites and spines, can be formed in the absence of presynaptic glutamate release[78]. Synaptic components may be trafficked to the surface before synapses are formed to allow rapid adaptation and mod-ification of cell-surface microdomains in response to external stimuli. For example, glutamate triggers de novo spine growth from the dendrite shaft via NMDA receptors within merely 20 s[79]. Similar results were also found in cells treated with GABA and $GABA_A$ receptors[80]. This rapid response suggests that pro-teins were present at the onset of stimulation.

Comparing the total pool with the surface abundance pool during neuronal development and synaptic plasticity, we inves-tigated the interplay of surface trafficking and total abundance change on a system-wide scale. We found that the surfaceome is more dynamic than the total proteome as indicated by the per-centage of proteins, that underwent significant changes during the 20 days of culture. When synaptic transmission was scaled as homeostatic mechanisms, we found that the fold-change differ-ence compared to untreated controls was significantly higher for the surface compared to the total abundance pool. Across the 20 days of culturing, surface and total levels for many proteins had high correlation, suggesting that the surface abundance is predominantly determined by global proteostasis. However, when we compared neighboring time points with 2 days between measurements, the correlation and overlap of significantly modulated surface proteins was decreased. These findings suggest that total protein abundance modulates the surfaceome on a broad scale over a long time period, whereas changes in surface trafficking are more pronounced at smaller time scales. The data also suggest that due to the level of regulation by surface traf-ficking, the surfaceome is more flexible and can achieve a higher dynamic range of protein quantities during a limited time window.

Furthermore, the uncoupling of surface and total abundance changes has functional implications. For synaptic vesicle mem-brane proteins, we detected the assembly of intracellular pools that increasingly became surface-exposed upon synaptic activity. We were able to dissect and identify differential surface traffick-ing with the autoCSC method; however, synaptic vesicle pools represent an extreme case in biology that exploits intracellular vesicle pools and tightly controlled exocytosis. We also identified differential regulation for the neuronal pentraxin receptor, potentially indicative of intracellular accumulation. Furthermore, our findings suggest a number of avenues for investigation in homeostatic plasticity. Notably, we observed overlap between cLTP treatment and synaptic scaling: Ptprn increased in surface abundance during synaptic downscaling and decreased abun-dance after cLTP.

Synaptic recruitment of AMPA receptors from extrasynaptic pools by surface diffusion is required for LTP[60]. Considering the

basal extrasynaptic reserve pool of AMPA receptors, the question has been raised whether exocytosis of new AMPA receptors is needed for early LTP or whether other cargo mediates the strict requirement of exocytosis for LTP[63,65–67]. Signaling mechanisms of synaptic plasticity differ in neuron cultures and hippocampal slices, therefore future studies are required to determine whether our findings from cLTP treatment of cultures are relevant for *bona fide* LTP. The data provided here from cLTP stimulation in vitro provides a number of avenues for further research. First, we identified numerous proteins with increased surface abundance upon cLTP treatment, and these factors could account for the necessity of exocytosis. Second, different synapses may produce different forms of plasticity and the multitude of effects, we observe in cortical cultures may not occur on the same cell or synapse but represent a mixture of events limited to specific synapse subtypes. For example, we observed a strong increase in abundance of AMPA auxiliary subunit stargazin, which has been shown to mediate synaptic trapping of AMPA receptors by direct interaction with PSD95[59]. Based on these results we can hypothesize that increased surface delivery of stargazin shifts the dynamic equilibrium between extrasynaptic and synaptic AMPA receptors to promote diffusional trapping of mobilized AMPA receptors, thereby connecting the requirement for exocytosis with surface diffusion and synaptic trapping.

## Methods

**Chemicals**. All chemicals were purchased from Sigma unless stated otherwise.

**Primary neuron culture**. Dissociated cortical neurons were prepared and maintained as previously described[81]. Briefly, cortices of postnatal day 0–1 rat pups (Sprague–Dawley) were dissociated and plated on poly-L-lysine-coated, 100-mm cell culture dishes (4–5 million cells per dish) or glass coverslips for immunocytochemistry. Cells were maintained in Neurobasal Plus media supplemented with B27 Plus and 2 mM Glutamax (all from Thermo Fisher Scientific) for the indicated number of days. Cytosine-D-arabinofuranoside (araC; 5 μM) was added to the medium on 4 DIV to prevent overgrowth of astrocytes. For microscopy LTP validation experiments, cortical cultures were prepared from E17 rat embryos as previously described[82]. For cLTP for autoCSC, the cell culture medium was removed, and cells were placed either in extracellular solution (120 mM NaCl, 3 mM KCl, 10 mM D-glucose, 10 mM HEPES, B27 Plus, 2 mM Glutamax, MEM amino acids, 2 mM MgSO₄, 2 mM CaCl₂, pH 7.4) or cLTP buffer (same as extracellular solution but with 0 mM MgSO₄, 3 mM CaCl₂, 30 μM bicuculline, and 200 μM glycine) for 5 min, followed by 20 min in extracellular solution before autoCSC surface labeling.

All animal experiments were carried out under institutional guidelines (ZH172/18 Kanton Zürich Gesundheitsdirektion Veterinäramt).

**Cell surface capture—labeling and digestion**. Surface glycoproteins on live cells were gently oxidized with 2 mM NaIO₄ (20 min, 4 °C) in labeling -buffer (LB) consisting of PBS, pH 6. Cells were washed once in LB and subsequently biotinylated in LB containing 5 mM biocytin hydrazide (Pitsch Nucleic Acids) and 5 mM 5-methoxyanthranilic acid for 1 h at 4 °C min. Experiments with homeostatic scaling and cLTP were performed using 2-(aminomethyl)benzimidazole (50 mM)[83] instead of 5-methoxyanthranilic acid as catalyst. Cells were washed five times with LB and harvested, lysed in lysis buffer (100 mM Tris, 1% sodium deoxycholate, 10 mM TCEP, 15 mM 2-chloroacetamide) by repeated sonication using a VialTweeter (Hielscher Ultrasonics), and heated to 95 °C for 5 min. Proteins were digested with LysC (Wako Chemicals) and trypsin overnight at 37 °C using an enzyme-to-protein ratio of 1:200. In order to inactivate trypsin and precipitate deoxycholate, samples were boiled for 20 min, acidified with 10% formic acid to approximately pH 3, and centrifuged 10 min at 16,000 × g. Peptide concentrations were determined in the supernatant using a NanoDrop 2000 instrument (Thermo Fisher Scientific) and normalized before aliquoting into a 96-well sample plate for automated processing. For total proteotype analysis, a fraction of the digested proteins was separated and desalted for liquid chromatography–tandem mass spectrometry (LC-MS/MS) analysis as described below.

**Automated processing**. A Versette liquid handling robot (Thermo Fisher Scientific) equipped with a Peltier element was used to adjust the temperature within 96-well plates during glycopeptide elution. Streptavidin tips were prepared by pushing a bottom filter membrane into disposable automation tips (Thermo Fisher Scientific). Each tip was filled with 80 μl of Pierce Streptavidin Plus UltraLink Resin (Thermo Fisher Scientific), and tips were sealed by compressing the resin with a

top filter membrane. Assembled tips were attached to the liquid handling robot and washed with 50 mM ammonium bicarbonate by repeated cycles of aspiration and dispensing (mixing). Biotinylated peptides were bound to the streptavidin resin over 2.5 h of mixing cycles. Subsequently the streptavidin tips were sequentially washed with 5 M NaCl, StimLys Buffer (100 mM NaCl, 100 mM glycerol, 50 mM Tris, 1% Triton X-100), with 50 mM ammonium bicarbonate, 100 mM NaHCO₃ (pH 11), and with 50 mM ammonium bicarbonate. For glycopeptide elution, streptavidin tips were incubated overnight in 50 mM ammonium bicarbonate containing 1000 units PNGase F (New England Biolabs) at 37 °C. The sample plate was then removed from the liquid handling robot and acidified to pH 2–3 with formic acid. Peptides were desalted with C18 UltraMicroSpin columns (The Nest Group) according to the manufacturer's instructions and dried in a SpeedVac concentrator (Thermo Fisher Scientific).

**LC–tandem mass spectrometry analysis**. For MS analysis, peptides were reconstituted in 5% acetonitrile and 0.1% formic-acid containing iRT peptides (Biognosys) and analyzed in DIA and DDA modes for spectral library generation. For spectral library generation, a fraction of the samples originating from the same condition were pooled to generate mixed pools for each condition. Peptides resulting from autoCSC were separated by reverse-phase chromatography on a high-pressure liquid chromatography (HPLC) column (75-μm inner diameter; New Objective) packed in-house with a 15-cm stationary phase ReproSil-Pur 120 A C18 1.9 μm (Dr. Maisch GmbH) and connected to an EASY-nLC 1000 instrument equipped with an autosampler (Thermo Fisher Scientific). The HPLC was coupled to a Q Exactive plus mass spectrometer equipped with a nanoelectrospray ion source (Thermo Fisher Scientific). Peptides were loaded onto the column with 100% buffer A (99% H₂O, 0.1% formic acid) and eluted at a constant flow rate of 300 nl/min with increasing buffer B (99.9% acetonitrile, 0.1% formic acid) over a nonlinear gradient. The DIA method, (Bruderer et al.[35]) contained 14 DIA segments of 35,000 resolution with IT set to auto, AGC of 3 × 10⁶, and a survey scan of 70,000 resolution with 120 ms max IT and AGC of 3 × 10⁶. The mass range was set to 350–1650 *m/z*. The default charge state was set to 2. Loop count 1 and normalized collision energy stepped at 25.5, 27.5, and 30. For the DDA, a TOP12 method was recorded with 60,000 resolution of the MS1 scan and 20 ms max IT and AGC of 3 × 10⁶. The MS2 scan was recorded with 70,000 resolution of the MS1 scan and 120 ms max IT and AGC of 3 × 10⁶. The covered mass range was identical to the DIA.

For total proteotype analysis, peptides were separated by reverse-phase chromatography on an HPLC column (75-μm inner diameter; New Objective) packed in-house with a 50-cm stationary phase ReproSil-Pur 120A C18 1.9 μm (Dr. Maisch GmbH) and connected to an EASY-nLC 1000 instrument equipped with an autosampler (Thermo Fisher Scientific). The HPLC was coupled to a Fusion mass spectrometer equipped with a nanoelectrospray ion source (Thermo Fisher Scientific). Peptides were loaded onto the column with 100% buffer A (99% H₂O, 0.1% formic acid) and eluted with increasing buffer B (99.9% acetonitrile, 0.1% formic acid) over a nonlinear gradient for 240 min. The DIA method[36] contained 26 DIA segments of 30,000 resolution with IT set to 60 ms, AGC of 3 × 10⁶, and a survey scan of 120,000 resolution with 60 ms max IT and AGC of 3 × 10⁶. The mass range was set to 350–1650 *m/z*. The default charge state was set to 2. Loop count 1 and normalized collision energy was stepped at 27. For the DDA, a 3-s cycle time method was recorded with 120,000 resolution of the MS1 scan and 20 ms max IT and AGC of 1 × 10⁶. The MS2 scan was recorded with 15,000 resolution of the MS1 scan and 120 ms max IT and AGC of 5 × 10⁴. The covered mass range was identical to the DIA. The mass spectrometry proteomics data have been deposited to the ProteomeXchange Consortium via the PRIDE[84] partner repository with the dataset identifier PXD014790 (user: reviewer14671@ebi.ac.uk, password: rNlMIesc).

**High-pH fractionation of peptides**. High-pH fractionation was performed as previously described[85]. Briefly, samples were resuspended in Buffer A (20 mM ammonium formate and 0.1% ammonia solution in water, pH 10) and 200 μg of sample was injected into an Agilent Infinity 1260 (HP Degasser, Vial Sampler, Cap Pump) and 1290 (Thermostat, FC-μS) system. The peptides were separated at 30 °C on a YMC-Triart C18 reversed-phase column with a diameter of 0.5 mm, length of 250 mm, particle size of 3 μm, and pore size of 12 nm. At a flow of 11 μl/min, the peptides were separated by a linear 56-min gradient from 5 to 35% Buffer B (20 mM ammonium formate, 0.1% ammonia solution, 90% acetonitrile in water, pH 10) against Buffer A (20 mM ammonium format, 0.1% ammonia solution, pH 10) followed by a linear 4-min gradient from 35 to 90% Buffer B against Buffer A and 6 min at 90% Buffer B. The resulting 36 fractions were pooled into 12 samples. The buffer of the pooled samples was evaporated using vacuum centrifugation at 45 °C.

**Data analysis of DIA LC-MS/MS**. LC-MS/MS DIA runs were analyzed with Spectronaut Pulsar X version 12 (Biognosys)[35,36] using default settings. Briefly, a spectral library was generated from pooled samples measured in DDA as described above. The collected DDA spectra were searched against UniprotKB (UniProt reference proteome including isoforms for *Rattus norvegicus*, retrieved September 2018) using the Sequest HT search engine within Thermo Proteome Discoverer version 2.1 (Thermo Fisher Scientific). We allowed up to two missed cleavages and

semispecific tryptic digestion. Carbamidomethylation was set as a fixed modification for cysteine, oxidation of methionine, and deamidation of arginine were set as variable modifications. Monoisotopic peptide tolerance was set to 10 ppm, and fragment mass tolerance was set to 0.02 Da. The identified proteins were assessed using Percolator and filtered using the high peptide confidence setting in Protein Discoverer. Analysis results were then imported to Spectronaut Pulsar version 12 (Biognosys) for the generation of spectral libraries.

Targeted data extraction of DIA-MS acquisitions was performed with Spectronaut version 12 (Biognosys AG) with default settings as previously described[35,36]. The proteotypic filter "only protein group specific" was applied. Extracted features were exported from Spectronaut using "Quantification Data Filtering" for statistical analysis with MSstats (version 3.8.6)[86] using default settings. Briefly, features were filtered for use for calculation of Protein Group Quantity as defined in Spectronaut settings, common contaminants were excluded. For autoCSC, the presence of consensus NXS/T sequence including a deamidation (+0.98 Da) at asparagine was required. For each protein, features were log-transformed and fitted to a mixed effect linear regression model for each sample in MSstats[86]. In MSstats, the model estimated fold change and statistical significance for all compared conditions. Significantly different proteins were determined by the threshold fold-change >1.5 and adjusted $p$-value <0.05. Benjamini–Hochberg method was used to account for multiple testing. In the comparison of all developmental time points with each other, $p$-value adjustment was performed on surface proteins with a fold-change >1.5. Protein abundance per sample or condition was used for further analysis and plotting. The neurodevelopmental time series autoCSC experiment was performed with three biological replicates for ten time points, one sample from four DIV was excluded due to technical error, generating 29 samples for final quantification for autoCSC. Homeostatic scaling autoCSC experiment was performed with seven biological replicates per condition, outliers were removed to generate 18 samples for final quantification. The cLTP autoCSC experiment was performed with twelve biological replicates per condition generating 24 samples for final quantification.

**Glycosylation site analysis**. For glycosylation site counting the following rules were followed: (i) only glycosylated peptides conforming to the NX[STC] consensus sequence were considered; (ii) to avoid inflating the count, nonproteotypic peptides were arbitrarily assigned to a single protein in the protein group; (iii) if a glycosylated peptide could be mapped to multiple positions within the same protein, both positions were kept, unless one of the mappings resulted in a higher number of sites matching the consensus NX[STC] motif, in which case only this one was kept. When comparing the glycosylation sites identified in this study to the ones annotated in UniProt, only proteins identified in this study as having at least one glycosylation site were considered in UniProt.

**Data analysis—correlation, clustering, and impulse model**. All further analysis was done based on quantitative values obtained from MSstats. Upset plots were generated using the group quantification table in UpsetR[87]. GO analysis of the neuronal surfaceome composition was performed with EnrichR[88]. Correlation analysis and significance testing was performed in MSstats and were calculated in R using standard functions. For fuzzy c-means clustering of surface abundance profiles we used mfuzz[37], and topGO was used for GO annotation of resulting clusters[89]. For comparison of surface and total abundance pools, quantitative values from MSstats were merged by protein groups. Protein groups were considered matching if at least one Uniprot protein entry of the surface protein group was found in a protein group from the total prototype analysis. Impulse modeling and significance testing were performed with ImpulseDE[47] using default settings for two time courses. As input, we used quantitative values for each condition (surface and total) from MSstats. Prior to ImpulseDE, both surface and total time series were centered around two DIV by subtracting the value of two DIV from each time point and ten was added to each data point to avoid negative values.

**Immunocytochemistry—synapse counts and astrocyte quantification**. Cells were washed in PBS and fixed in 4% PFA for 15 min at room temperature. After washing in PBS, cells were permeabilized with 0.25% Triton X-100 in PBS for 5 min and blocked with PBS containing 10% BSA for 30 min at 37 °C. Cultures were stained with primary antibodies against PSD95 (mouse monoclonal, clone 6G6-1C9, Thermo Fisher Scientific, 1:1000), gephyrin (recombinant rabbit purified IgG, clone RbmAb7a, Synaptic Systems, 1:500), map2 (rabbit polyclonal, ab32454, abcam, 1:500) or GFAP (mouse monoclonal, Cell Signaling 3670, 1:500) in PBS containing 5% BSA overnight at room temperature. After washing, cells were incubated with secondary antibodies (antimouse IgG Alexa Fluor 555, Molecular Probes A21424, 1:500 and donkey antirabbit IgG H&L Alexa Fluor 647, Abcam ab150075, 1:500) for 45 min at room temperature. For synapse counts, cells were incubated additionally with primary antibody against synapsin (monoclonal mouse IgG fluorescently labeled with Oyster 488, clone 46.1, Synaptic Systems, 1:1000) in PBS containing 5% BSA for 4 h. Cells were washed again in PBS and mounted on slides in Prolong Gold Antifade (Thermo Fisher Scientific). Images were acquired using a 63 × 1.4 NA Oil Plan-Apochromat DIC M27 objective on a Zeiss LSM 880 upright laser scanning confocal microscope. Z-stacks of ten random positions on 3–4 different coverslips were taken per DIV for synapse quantification. Z-stacks of

four random positions on eight different coverslips were taken per cell type quantification. Maximum intensity projections were analyzed in ImageJ, and the Synapse Counter plugin was used with default settings[90].

**Immunocytochemistry—live cell labeling validation experiments**. cLTP was performed on DIV 14 neuron cultures plated on glass coverslips via coverslip transfer from home media to either blocking solution (control; 117 mM NaCl, 5.3 mM KCl, 30 mM D-glucose, 26 mM NaH$_2$CO$_3$, 1 mM NaH$_2$PO$_4$, 15 mM HEPES, B27, 2 mM Glutamax, MEM amino acids, 2 mM MgSO$_4$, 2 mM CaCl$_2$) or cLTP solution (same as control but with 0 mM MgSO$_4$, 3 mM CaCl$_2$, 30 μM bicuculline, 200 μM glycine) for 5 min, followed by 20 min recovery in blocking solution before live immunostaining. For bicuculline and TTX experiments, drug or vehicle was delivered to cells 24 h prior to surface labeling. Antibodies targeting the extracellular domains of ADGRB1 (ABR-021, Alomone), AMPARs (182 411, Synaptic Systems), and AC3 (AAR-043, Alomone) were diluted 1:100 in blocking solution containing 10% normal goat serum (NGS) and incubated with the cells for 60 min. Previously characterized homemade antibodies[91] targeting the extracellular domain of GABA$_A$ receptors (guineapig antiGABRG2, guineapig antiGABRA1, and guineapig antiGABRA2) were diluted at 1:400. Cells were washed 3× in PBS before live staining at 4 °C for 60 min, then washed 3× in PBS before fixation in 4% PFA. Cells were permeabilized with 0.1% Triton in PBS with 10% NGS prior to intracellular labeling of synapsin (106 004, Synaptic Systems) at 1:1000 for 90 min at room temperature. Cells were washed 3× in PBS before labeling with secondary antibodies conjugated to Alexa 647, Alexa 488, or Cy3 (Jackson Immuno Research) for 30 min at room temperature, then washed 3× with PBS. Imaging was performed using a LSM800 confocal microscope (Zeiss) with a 40× objective (NA 1.4) at 16 bit depth under nonsaturating conditions using a step size of 0.5 μm for a total of six z-stacks. Image analysis was performed on the proximal section of principal dendrites (each sample represents one neuron). For total intensity measurements ImageJ was used to quantify the summed intensity across all z-stacks. For synaptic puncta analysis, a custom in house ImageJ macro described previously[82] was used to quantify the number of puncta per unit area (density) as well as puncta size and integrated density. Mean differences between control and cLTP conditions were analysed using either a two-way $t$-test (for normally distributed data) or a Mann–Whitney test (for non-normally distributed data) using Prism 5 (GraphPad).

**In situ hybridization**. In situ hybridization was performed using the QuantiGene ViewRNA kit from Panomics as previously described[92]. Briefly, cells were fixed for 30 min at room temperature using a 4% paraformaldehyde solution (4% paraformaldehyde, 5.4% Glucose, 0.01 M sodium metaperiodate, in lysine-phosphate buffer). After completion of in situ hybridizations using probe sets against GAD1 and GAD2, cells were washed with PBS 1× and incubated in blocking buffer (4% goat serum in PBS 1×) for 1 h, followed by incubation with the respective antibodies for cell type markers (MAP2 and GFAP), washed in PBS and incubation with fluorophore-coupled secondary antibodies for 30 min. Cells were washed, counterstained with DAPI in PBS (Roth, 1 μg/ml) for 3 min and imaged directly in PBS. Images were acquired using a 20× objective on a Zeiss LSM 780 laser scanning confocal microscope. Z-stacks of three random positions on nine different coverslips were taken. Maximum intensity projections were analyzed using ImageJ.

**Electrophysiology**. For electrophysiological experiments, cortical culture neurons were plated on a glass coverslip at a density of 40 k and patch-clamp recordings were made on DIV 14–16.

For homeostatic scaling experiments, cells were stimulated for 24 h with 40 μM bicuculline (BIC) or 1 μM tetrodotoxin (TTX) at 37 °C. Control cells were not treated. For cLTP experiments, the coverslips with the cells were incubated in the cLTP solution (120 nM NaCl, 3 mM KCl, 10 mM D-Glucose, 10 mM HEPES, 1 mM L-glutamine, 0 mM MgSO$_4$, 3 mM CaCl$_2$, 200 μM glycine, 20 μM bicuculline) at 37 °C for 5 min and subsequently for 15 min at 37 °C in recording solution (120 nM NaCl, 3 mM KCl, 10 mM D-Glucose, 10 mM HEPES, 1 mM L-glutamine, 2 mM MgSO$_4$, 2 mM CaCl$_2$, 0 μM glycine, 0 μM bicuculline). Control cells were incubated for 15 min at 37 °C in recording solution. For homeostatic scaling experiments, the same recording solution was used.

For patch-clamp recordings, the cells were visualized by infrared differential interference contrast optics in an upright microscope (Olympus BX-51WI) using a Hamamatsu Orca-Flash 4.0 CMOS camera. Recordings were performed using borosilicate glass pipettes with filament (Harvard Apparatus; GC150F10; o.d., 1.5 mm; i.d., 0.86 mm; 10-cm length) at 33 °C with an intracellular solution of 95 mM K-gluconate, 50 mM KCl, 10 mM Hepes, 4 mM Mg-ATP, 0.5 Na-GTP, 10 mM phosphocreatine; pH 7.2, KOH adjusted, 296 mOsm.

Electrophysiological recordings were made using a MultiClamp700B amplifier (Molecular Devices), and signals were filtered at 10 kHz (Bessel filter) and digitized (50 kHz) with a Digidata1440A and pClamp10 (Molecular Devices). Spontaneous events were recorded in voltage clamp mode at −60 mV for 5 min. The data analysis was performed using Clampfit (Molecular Devices). In each cell, spontaneous synaptic events were selected by an experimenter who was blinded to the applied treatment. As a proxy for excitatory postsynaptic events (EPSCs), events with half-width <3 ms and decay time constant <6.5 ms were considered.

For final analysis, the mean amplitude of fast EPSCs were considered for each cell (see EPSC amplitude plots). Cells without any detected fast EPSC were not considered in the amplitude comparisons. For determining burst frequencies, the number of bursts were counted in each cell and divided by the recording time.

**Reporting summary**. Further information on research design is available in the Nature Research Reporting Summary linked to this article.

## Data availability

The mass spectrometry proteomics data have been deposited to the ProteomeXchange Consortium via the PRIDE[84] partner repository with the dataset identifier PXD014790. Selected data is available at http://neurosurfaceome.ethz.ch/. Source data are provided with this paper.

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

## Acknowledgements

We are grateful to the members of the Wollscheid and Aebersold research groups for discussion and support at all stages of the project. We thank M. Stamou for teaching and sharing experimental procedures; J. R. Wyatt for editing; T. Splettstoeser for graphical support; R. Aebersold, I. Mansuy and E. Schuman for discussions and sharing of experimental procedures; M. Kopf, J. Kieselow, A. Alitalo, and EPIC staff for support with animal experimentation; and T. Schwarz and ScopeM for support with microscopy. For funding, we acknowledge the ETH (grant ETH-30 17-1 and grant ETH-25 15-2) and the Novartis Foundation for Biomedical Research and the Swiss National Science Foundation (grant 31003A_160259 for B.W.).

## Author contributions

M.v.O. performed all experiments except those noted below and wrote the manuscript. M.M. contributed new analytical tools. P.G.A.P. performed glycosylation site analysis. B.C. and S.K.T. performed live cell labeling validation experiments. C.S. and C.F. performed electrophysiology experiments. M.v.O. and J.H. performed cell type characterization. M.v.O. and S.t.D. performed FISH. M.M. and P.G.A.P. developed the R shiny webpage. B.W. supervised the project and wrote the manuscript. All authors edited the manuscript.

## Competing interests

The authors declare no competing interests.
