## [Peer Review File · Nature Communications]

Reviewers' comments:

Reviewer #1 (Remarks to the Author):

Van Oostrum et al. analyze the surface proteome of cultured cortical neurons during their development in vitro, and following induction of homeostatic plasticity and chemical LTP. During development in culture, the authors identify 6 different clusters of proteins with distinct developmental dynamics. Synaptic proteins are already present at the surface before synaptogenesis occurs. The authors then manipulate activity in their cultures with various treatments designed to induce synaptic scaling or synaptic potentiation. They observe a dynamic reorganization of the neuronal surface during homeostatic plasticity and cLTP. The surface protein Ptpn is regulated during both manipulations. Overall, the study provides an overview of the dynamic changes in surface proteome composition during neuronal development in culture and following changes in synaptic activity. This is useful and can serve as a resource for others. On the other hand, no major new mechanistic insight emerges from this work. Much of the data remains descriptive without validation.

Major points:

1. Many things change during DIV2 and DIV20 in primary cultures that could influence the quantification of CSPs during development (volume to surface ratio, glia to neuron ratio) that cannot be normalized by total protein content prior to N-glycopeptide capture. How do the authors control for these changes?
2. The authors nicely validate the increased surface expression of some proteins upon cLTP. Similar validation experiments should be performed for changes in surface expression during neuronal development and homeostatic plasticity. Is it possible that extracellular N-glycosylation is changed instead of actual protein expression?
3. It is surprising that 2 out of 6 clusters (clusters 3 and 6) (Fig.2) are enriched for proteins associated with intracellular membranes considering that the authors' strategy is designed to capture surface proteins. A check in Uniprot of several of the proteins in supplementary table 1 yields multiple proteins that are thought to reside inside the cell. While it is conceivable that some of these may become surface-exposed at some point, 33% of clusters seems high. The authors should perform validation experiments to check that they are not detecting intracellular proteins.
4. Synapses represent a small fraction of the neuronal surface, but the synaptic activity manipulations in Fig.5 (homeostatic plasticity) and Fig.6 (cLTP) will primarily affect the synaptic surface. Their surfaceome analysis may therefore underestimate local changes at the synapse.
5. The scaling experiment in Fig.5 and the cLTP experiment in Fig.6 lack validation to show that the manipulations were effective. Experimental validation should be provided. This is especially relevant

for the scaling experiment, where the authors used 24hr treatment instead of the more commonly used 48hrs.

6. Experimental validation of some of the findings is required to determine functional relevance of these findings. An interesting target protein the authors could focus on is Ptpn, which the authors show is regulated in both scaling and cLTP. Perturbation of Ptpn expression levels in combination with manipulation of activity can determine whether Ptpn is functionally required in scaling and cLTP .

Minor points:

1. How fast is extracellular N-glycosylation and does this bias the authors towards the identification of more stable CSPs?

2. The phrasing of the authors makes it seem like all synapse assembly factors and synaptic proteins are trafficked to the surface prior to synapse formation due to enrichment of these annotations in certain clusters. However, this does not mean that all of them behave this way and in fact this is apparent in Fig.3b and S8a.

3. The conclusion on page 9 that homeostatic scaling predominantly influences the postsynapse is at odds with the increase in surface SV2a/b and thus synaptic vesicle fusion (a presynaptic effect) the authors show. Please explain.

4. The authors claim that the surface/global proteome correlation is dependent on the time between measurements. Is this not influenced by the opposite change in correlation in the surface proteome between measurements as seen in fig 1C?

5. Page 12: increased surface abundance does not necessarily equal increased exocytosis. Please rephrase.

6. As a suggestion, synapse development in neuronal cultures is well established and the first paragraph of the results describing this could be condensed in order to start with novel findings that capture the attention of the reader.

Reviewer #2 (Remarks to the Author):

Here van Oostrum et al., investigated the temporally-resolved surfaceome analysis of developing primary neuronal cultures. The study offers an interesting resource for the analysis of membrane proteins in primary neuronal cultures. However, the assays used are a combination of in-vitro and non-physiological conditions that makes difficult to extrapolate the results and conclusions, in particular to different forms of synaptic plasticity. As a resource manuscript it fails to provide to the reader with a clear and ease to use dataset.

I found extremely difficult to follow and evaluate the observations and conclusions of the manuscript since the tables provides only uniprot identifiers without indicating gene or protein id. While uniprot ids can be converted to any other identifier, the authors need to provide all the information in an appropriate format not just for the reviewers but for the readers.

While I cannot comment on the findings and conclusions of the manuscript without further evaluation of the molecules described, my main concern is on the utility of the dataset, in particular regarding different forms of synaptic plasticity. While the results can be of use within the context of neuronal cultures, the manuscript extrapolate the findings to synaptic plasticity. It is true that the glycine chem-LTP protocol have been extensively used in neuronal cultures, however these in vitro neurons do not show LTP and it is very well known that signaling mechanisms differ in cultured neurons and hippocampus slices. The authors claim: "Primary neuronal cultures have been used to study fundamental aspects of neuronal development, synapse formation, and synaptic plasticity" While this is true, it doesn't mean that this is the method of choice, or that our understanding of synaptic plasticity mechanisms are due to analysis in cell culture systems. Generally neuronal cultures are not the method of choice or most indicated to study synaptic plasticity. While insertion of AMPAR are necessary for LTP, measurement of this parameter of neuronal cultures is not a measure of LTP. Ideally, the assays should have been performed in slice preparations.

A way forward might be to discuss the results within the context on an in-vitro system, which can produce chem-LTP in slices, without firmly stating that the results obtained underlie LTP.

Another option is to show that the chem-LTP protocol can induce evoked excitatory postsynaptic currents (EPSCs) at synaptic connections between individual hippocampal neurons. This can be done using paired whole-cell recordings from pyramidal neurons. Then results can be discussed within this context, but not LTP.

A similar scenario is observed for homeostatic plasticity. Here, it is more complicated to address the results within the context of neuronal physiological processes. One, is the inherent problems to address the temporal scales of different forms of plasticity and second, the differences observed in culture and slice models.

While silencing neuronal activity with TTX has been extensively used to address processes involved with homeostatic plasticity, Hebbian and homeostatic forms of synaptic plasticity operate on vastly different time scales. Thus the field have been always struggled to explain how forms of homeostatic plasticity that develop over hours and days (here cultures are incubated by 24hs with TTX or Bic) can provide the negative feedback needed to regulate synaptic weights and stabilize activity in cells and circuits undergoing Hebbian synaptic plasticity, which can be induced in minutes or seconds (see chem LTP protocol used). The use of cell culture models makes more difficult to interpret within a physiological context. It is also known that homeostatic plasticity is differentially regulated under development. There is no rationale on the developmental stage selected or how this compares to a slice model. Both scenarios can be summarized in that TTX has no effect on mini frequency in cultured neurons (14 DIV or less), while the in-vivo infusion of TTX into the hippocampus of adult

rodents does induce a profound increase in mini frequency while no effect on mini amplitude in CA1 pyramidal cells. Irrespective of performing the assays in hippocampal slice or neuronal cultures. The protocols selected are of difficult comparison, by harshly blocking neuronal activity or mildly modulating GABAergic/Glutamate signaling. Again the physiological importance of blocking electrical activity for 24hs as a measure of homeostatic plasticity in a cell culture system is questionable. Again, it will be nice to have a physiological correlate to compare, more mechanistic, other than AMPAR insertion.

Other:

The authors indicate that: on average 72% of synapses were excitatory. Does this mean that they determined a 28% of inhibitory synapses in their cultures? The number seems a bit high. How this number was calculated? What is the percentage of inhibitory neurons in the culture? Any bias toward somatostatin, pv+ neurons? Numbers are highly variable between culture conditions. There is no report of inhibitory neurons in the manuscript. This is relevant in particular when considering the Bic protocol.

There are a number of statements that are not clear or overstated, and the analysis do not show or help to have an in-depth understanding of the organization of surface proteins, for example:

“This analysis revealed several subnetworks. For example, the ephrin family is primarily associated with clusters 4 and 5 and cadherins are mostly associated with clusters 1 and 2. Cell adhesion molecules are relatively underrepresented in clusters 3 and 6, which involve proteins that increase in abundance starting at 12 DIV. This supports the notion that surface expression of cell adhesion molecules does not increase during synapse formation or synaptic activity”.

In general protein functions and families are overlooked and discussed within the context of trends in quantitative changes at different developmental stages. Together with the fact that only uniprot ids are provided, this makes the reading complicated and the conclusions difficult to evaluate.

For example, the authors say:

mRNAs that encode cluster 4 and 5 proteins are enriched in astrocytes

however, they said before:

“For example, the ephrin family is primarily associated with clusters 4 and 5” and they conclude: This supports the notion that surface expression of cell adhesion molecules does not increase during synapse formation or synaptic activity.

The, the authors are now discussing ephrins (which are expressed also in neurons) together with a cluster enriched in astrocytes. This might be ok from the point of view of clustering molecules with a particular criteria, but the conclusions not only are very confusing for the general reader, but also can be very misleading depending on the considerations for clustering analysis. Is not clear if the clustering analysis consider just trends in protein levels or also cell types. The cultures have at least three main cell types, excitatory, inhibitory neurons and astrocytes. A relative quantitation for each cell type will be nice.

The authors said: The total abundance is determined by the interplay of protein synthesis and degradation, and regulated surface trafficking likely accounts for the higher extent of differential expression on the surface. It is not clear if the authors considered or normalized their results against the number of synapses present.

Suggesting neuronal firing events lead to a net increase in synaptic vesicle fusion events. These results agree with previous findings that homeostatic synaptic scaling predominantly influences the postsynapse rather than presynaptic vesicle release and recycling. It is not clear why synaptic vesicle fusion events inform on postsynapse rather than presynaptic vesicle release

It is not clear what the authors means by:

Furthermore, these results indicate that there are relatively few qualitative alterations to the surfaceome during

maturation in culture, suggesting that regulation rather affects the quantitative surface abundance.

we found that Gria1 and Gria2 had significantly increased surface abundance upon stimulation (Fig. 6b, c) indicating that the cLTP protocol triggers AMPA receptor exocytosis.

This is extensively reported in multiple systems including chem LTP in neuronal cultures

The strongest responder was the receptor type tyrosine-protein phosphatase Ptpn.

Here it is not clear if Ptpn is discussed within the context of postsynaptic or presynaptic membrane. This is important because of pre/postsynaptic site in synaptic plasticity. PTPR proteins have been

usually described at the presynaptic site. It is not clear if the authors are claiming a role for Ptpn at the postsynaptic site in relation to AMPAR trafficking. If so, it will be good to show that Ptpn is present and have a role at the postsynaptic site or if the observed correlation is to the presynaptic site. In fact this might occur with a number of the reported proteins, without having a knowledge of their localization. I acknowledge that this is not easy to perform, but also the statements should be more careful and in line with the assays performed.

The neuronal surfaceome interaction network.

There are no major details on how the interaction network was built. Is this a protein-protein interaction (PPI) network? How the information was curated. PPI networks usually include all sorts of information, obtained by different methods (Y2H, IP, overexpression, etc.), developmental stages, different species, cell types, etc... Is extremely important to perform a manual curation of the data used. In particular for membrane proteins where IP assays usually include "pieces" of membrane with all sorts of non-interacting membrane proteins.

The interaction network figures are not clear I don't know what the edges of the network means, or which ones are the nodes connected. This info might be within the uniport ids but it is not clear. For example, does the figure implies a PPI between Grm1, Gria1? Where this info is coming from? Or between Unc5a/d-Dcc/Dscam? Nrnx2-Nrnx3 (these are just some examples) there are many edges that are unknown PPI.

The decrease of Nlgn1 seems strange as it has been reported to increase through development in excitatory synapses. Are there any other Nlgn family members increasing through neuronal development? Or any comments on this?. Again it is difficult to follow the data with just a figure

Methods:

What's the reasoning behind using bicuculine in a Mg free solution with 200uM Gly?

What's the rationale behind performing c- LTP assays at DIV 14 and not on DIV 20?

Reviewer #3 (Remarks to the Author):

The MS by Oostrum et al. reports the application of a previously published proteomic method for quantifying cell surface proteins to neuronal cell cultures. In particular, the authors investigate changes in the cell surface composition during neuronal development (in culture), and synaptic plasticity experiments. While many of the observed changes correlate with overall changes in protein abundance, they identify notable exceptions, ie proteins whose cell surface abundance appears to be strongly regulated by trafficking. The MS thus reports on a generally useful and scalable experimental approach to study the neuronal 'surfaceome', and provides good sample applications providing new insights into neuronal surface dynamics.

The MS is well written, the experiments are technically sound, and the results are well presented. The approach is timely and powerful. Nevertheless, I would like to raise two important issues, that currently limit the impact of the study.

1. The MS repeatedly states that the method captures only 'bona fide' cell surface proteins, and no intracellular proteins. However, as discussed further below, a substantial proportion of intracellular proteins are also recovered, many of them from the endoplasmic reticulum and from lysosomes. Using the provided data and external subcellular localisation annotation, I estimate the 'purity' at around 80% plasma membrane proteins – good, but by no means pure. Hence, any claims to 100% specificity need to be toned down considerably, and all findings in the study need to be re-evaluated in light of the reduced specificity, where necessary.

2. The study uses cultured cortical neurons, which are in fact a mixture of astrocytes and neurons. This very important point is mentioned rather late in the results section, but has major implications for the entire study. Without orthogonal information, it is not possible to decide which of the identified proteins are neuronal and which ones are from astrocytes. But this is critical for any inferences regarding the neuronal surface. No attempt was made to quantify the relative proportions of astrocytes and neurons. Importantly, the proportions might change during the course of the differentiation, and this may well account for some of the observed changes. The authors need to address this point in detail. The short foray into gene expression (Supp. Figure 8) is in my opinion insufficient.

Since the proposed approach has the potential to become a widely used 'gold-standard' for investigating neuronal surface dynamics, it will be very important to discuss and address these limitations.

Specific comments

The authors sub-cluster the 943 identified 'surface' proteins into six groups, based on common changes during the time course (Fig. 2). They claim that all of these proteins are 'bona fide' cell surface proteins.

I have used Uniprot subcellular localisation annotation for rat proteins to annotate the six clusters for a rough compositional analysis. Uniprot provided localisation annotations for 381 of the 943 proteins. Of these 30 were annotated as ER, 25 as Golgi, 17 as lysosomal, 4 as nuclear, 1 as mitochondrial – so in sum $77/381 = 20\%$ with a clear non-plasma membrane localisation. So even assuming that all other annotated proteins are indeed plasma membrane localised, one would estimate the proportion of PM proteins at around 80%.

This analysis is also in line with the estimates of PM purity by Weekes et al. (2010, *J Biomol Tech.* 21:108-115), who compared different cell surface labelling and capture approaches, including the present one (59-85% estimated purity).

Please note that I do not wish to argue the exact level of purity here; the important point is that there is a substantial pool of intracellular proteins, and this needs to be considered.

The proportion of plasma membrane proteins varies widely between clusters. Clusters 1, 2, 4, and 5 are highly enriched in plasma membrane proteins, but 3 and 6 contain only few. Indeed, the authors remark on this, and try to explain it:

"Clusters 3 and 6 share a curious feature: Both show an acute increase in abundance starting at 12 DIV (Fig. 2a), the time point when synapse formation increases considerably (Supplementary Fig. 1b). Interestingly, both clusters are enriched in proteins associated with intracellular membranes, but the two clusters are enriched in membranes from different organelles (cluster 3: "endoplasmic reticulum part", "membrane enclosed lumen", cluster 6: "vacuolar membrane", "lysosomal membrane", "coated vesicle"). These intracellular membranes potentially appear at the surface at

increased rates due to alterations in surface trafficking. Associated annotations point to functions related to degradation or clearing of biomolecules from the surface (cluster 3: "proteolysis in cellular catabolic process", "ERAD Pathway", cluster 6: "catabolic process", "autophagy", "hydrolase activity") but also synthesis of new biomolecules (cluster 3: "glycoprotein biosynthetic process", "macromolecules biosynthetic process") (Fig. 2b). These findings could reflect acute changes in the dynamic interplay of biosynthesis, endosomal degradation, recycling, and surface trafficking, all important for control of synapse formation and modulation of synaptic plasticity."

While it is not impossible for an integral membrane protein of the ER to appear at the cell surface transiently, major steady state pools of ER proteins at the cell surface are, to my knowledge, not a common occurrence (as the data presented here would suggest); how a luminal ER protein would be retained at the cell surface would also require an explanation. And what would be the function of trafficking biosynthetic ER proteins or a nuclear pore complex member (NUP210) to the cell surface? A much simpler explanation is either a limited unwanted permeation of the cell membrane by the labelling reagent, resulting in labelling of intracellular proteins; or perhaps the labelling of cell debris/dead cell remnants in the cell culture medium.

So for a proper interpretation of the data in the context of neuronal development, I think it will be necessary to acknowledge the labelling of some non-surface proteins, and curate the list of detected proteins accordingly.

Related to this problem:

P6/7

"Proteins from clusters 3 and 6 are encoded by mRNAs more highly expressed in neurons than in astrocytes, whereas mRNAs that encode cluster 4 and 5 proteins are enriched in astrocytes."

In light of my comments above, I find this rather alarming – the most neuron-specific proteins are in the clusters that contain the smallest proportion of surface proteins. So their behaviour will not inform on cell surface processes.

P5

How does Supp. Figure 6 show that 6 is the optimum number of clusters? To me it looks like the minimum of the graph is at cluster size of 8 or 9? Please clarify.

P8 and Figure 4C

"Together, this demonstrates that for many proteins the total abundance pool correlates with the surface pool over the entire time course of neuronal differentiation in culture but there are considerable fluctuations over time. These findings suggest that changes in surface trafficking influence surface abundance generally on short time scales, whereas total protein abundance modulates the surfaceome over long time periods."

A simpler explanation is that cultures with close time points will be more similar (as Figure 1C shows), and hence there will be fewer genuine differences, and likely of smaller magnitude. These will therefore also not correlate so well. But if you look at the most distant time points, the differences will be largest, and hence easier to quantify accurately, resulting in better correlation.

To test this, the authors could for example compare the median magnitude of 'significant' differences between adjacent time points – is this smaller than the median difference for the most distant time points?

Table S2

I have tried and failed to extract the 128 proteins that show significant differences between surface and proteome pool changes from Table S2. Please provide these in a separate list, with gene names annotated, to make this a more useful resource.

Fig 6i

Please show a control of a protein not changing to make this figure more convincing. See also my next comment.

Fig 6i, legend

Two different statistical tests were used for different proteins in the same experiment – this is highly unusual and will need to be justified.

Discussion:

"Here, we present an initial blueprint of how many, and which, proteins are present on the surface of neurons. First, neurons come in different populations and our rat cortical neuronal culture system contains a mixture of these cells. Furthermore, our cultures contain predominantly neurons, but also astrocytes. We addressed this question partially by use of transcriptomic data; a more complete delineation could be achieved by combination with cell-type specific metabolic labeling approaches."

See my major point 2 above. Since no attempts were made to quantify the neuronal and astrocyte populations, any conclusions regarding cell surface dynamics in neurons are limited.

Point-by-point Response NCOMMS-19-34283A

Reviewer #1 (Remarks to the Author):

Van Oostrum et al. analyze the surface proteome of cultured cortical neurons during their development in vitro, and following induction of homeostatic plasticity and chemical LTP. During development in culture, the authors identify 6 different clusters of proteins with distinct developmental dynamics. Synaptic proteins are already present at the surface before synaptogenesis occurs. The authors then manipulate activity in their cultures with various treatments designed to induce synaptic scaling or synaptic potentiation. They observe a dynamic reorganization of the neuronal surface during homeostatic plasticity and cLTP. The surface protein Ptprn is regulated during both manipulations. Overall, the study provides an overview of the dynamic changes in surface proteome composition during neuronal development in culture and following changes in synaptic activity. This is useful and can serve as a resource for others. On the other hand, no major new mechanistic insight emerges from this work. Much of the data remains descriptive without validation.

We would like to thank reviewer 1 for the concise and good summary of our research presented in this manuscript.

Major points:

1. Many things change during DIV2 and DIV20 in primary cultures that could influence the quantification of CSPs during development (volume to surface ratio, glia to neuron ratio) that cannot be normalized by total protein content prior to N-glycopeptide capture. How do the authors control for these changes?

The data of our experiment comparing the surfaceome during development from DIV2 to DIV20 is normalized on peptide level prior to N-glycopeptide capture and also bioinformatically by median equalization within the statistical framework of MSstats. As such our data provides relative quantification of each surface protein across different timepoints normalized to total protein abundance.

Reviewer 1 raised an interesting point, what about the volume to surface ratio? Morphologically there are substantial changes during development which will affect the cell volume and also the surface area. Incorporation of such changes would enable analysis of surface protein densities, i.e. the abundance relative to the surface area. Such information cannot be derived from our data and it is conceivable that the surface density for a particular protein remains constant when the surface area increases concurrently with the protein abundance. However, incorporation of such considerations into surface proteotype analysis is not trivial and we are not aware of an approach that successfully demonstrated such analysis. Furthermore, the distribution of many surface proteins is not random but highly organized for example at synapses where proteins cluster at high density with controlled nanoscale organization. Therefore, the benefit of averaged surface protein density information is currently unclear and we believe that this is beyond the scope of this manuscript.

Additionally Reviewer 1 asked about the glia to neuron ratio. The cortical cultures used here are primary cultures and as such they may contain any cell type present in the tissue of origin, which is predominantly neurons but also astrocytes. Ratios between cell types do not change during development because neurons are postmitotic and proliferation of glia is inhibited by addition of cytosine arabinoside (AraC) on (see methods). Because the treatment is at DIV4, there might be a bias in the first time point (DIV2). In order to highlight this caveat, we added this information to the main text. Additionally we conducted experiments to characterize the composition of the culture, we added two panels to supplementary figure 1 and updated the main text with the following information:

- New supplementary Fig. 1 panels c and d
- “To further characterize our cortical culture system, we established the relative abundances of neurons and astrocytes. Of all cells that could be categorized based on Gfap and Map2 immunofluorescence, we found an average of 81% to be neurons (**Supplementary Fig. 1c-d**). Of those, we classified 23% on average as inhibitory neurons based on the presence of GAD1/2 mRNA at the cell soma (**Supplementary Fig. 1c-d**).”
- “In order to preclude glial overgrowth, the cultures were treated with cytosine arabinoside (AraC) on DIV4.”

2. The authors nicely validate the increased surface expression of some proteins upon cLTP. Similar validation experiments should be performed for changes in surface expression during neuronal development and homeostatic plasticity. Is it possible that extracellular N-glycosylation is changed instead of actual protein expression?

Reviewer 1 raises an important point, is it possible that extracellular N-glycosylation is changed instead of protein surface abundance? As mentioned in the discussion, our cell surface proteotyping strategy is limited to proteins with extracellular N-glycosylation, which is a common feature among surface proteins with more than 95% of cell surface proteins having an extracellular glycosylation motif. Additionally, N-glycosylation aids folding in the secretory pathway and plays a role in protein quality control. Currently it is unclear for most glycoproteins whether unglycosylated proteoforms exist and can potentially bypass quality control to reach the cell surface. Nevertheless, it is an inherent limitation of the glyco-Cell Surface Capture (glyco-CSC) that potential changes in site occupancy cannot be distinguished from changes in glycopeptide abundance. In order to investigate this, we searched the total proteome dataset for presence of unglycosylated peptide forms that match the identified (de-)glycosylated peptides from CSC. In the data from neuronal development, we could merely identify 3 peptides out of potential >1'900, which is well below the 1% FDR rate for this experiment with a depth of >30'000 unique protein-group specific quantified peptides.

However, such investigations are only indicative and the best strategy is validation using glycosylation-independent methods. We thank the reviewer for acknowledging our efforts in validating the increased surface expression of selected proteins upon cLTP using neuro-morphology. The difficulty with these experiments is the availability of high quality antibodies that target the extracellular domain and are suitable for live-cell labeling. For proteins with such antibodies available, we performed additional validation experiments for cLTP, homeostatic plasticity and neuronal development. We added new Supplementary Figures 6, 15, 17b-c and updated the main text with the following information:

- New Supplementary Fig. 6: Validation experiments for neuronal development
- New Supplementary Fig. 15: Validation experiments for homeostatic plasticity

- New Supplementary Fig. 17b-e: Validation experiments for cLTP
- “In order to validate the results from autoCSC using an orthogonal methodology we performed live-cell surface labeling for GABA and AMPA receptors at DIV6 and DIV16 and could confirm the significant increase in surface abundance with immunocytochemistry (**Supplementary Fig. 6**).”
- “In order to validate the results from autoCSC using an orthogonal methodology we performed live-cell surface labeling followed by immunocytochemistry for GABA_A receptor subunits gamma-2 and alpha-1. For the gamma-2 subunit, we could confirm the significant increase in surface abundance upon both synaptic up- and downscaling. For the GABA_A receptor subunit alpha-1 we found no significant difference by immunocytochemistry, in line with our results from autoCSC (**Supplementary Fig. 15**).”
- “Additionally we could confirm for two subunits of the GABA_A receptor (GABRA1 and GABRA2) that they do not change significantly upon cLTP (**Supplementary Fig. 17b-e**).”

While information on protein surface abundance changes are scarcely available for neuronal development and neuronal plasticity, we additionally provide in the main text a selection of examples where our data can replicate previous studies using orthogonal methods.

- In agreement with our data, it was previously reported that both the surface abundance and total abundance of Nptx1 increases after 24 hours of TTX treatment (Immunocytochemistry for surface abundance, and Western blot for total abundance, Schaukowitch et. al. 2017)
- Diering et al. found no significant changes for Gria1, Gria2, or Gria3 surface abundance after 24 hours of TTX treatment, in agreement with our findings. (surface biotinylation followed by Western Blot, Diering et. al. 2014)
- The increase in surface AMPA receptors after cLTP stimulation is widely established as a hallmark of LTP (Reviewed for example in Nicoll, 2017)

3. It is surprising that 2 out of 6 clusters (clusters 3 and 6) (Fig.2) are enriched for proteins associated with intracellular membranes considering that the authors' strategy is designed to capture surface proteins. A check in Uniprot of several of the proteins in supplementary table 1 yields multiple proteins that are thought to reside inside the cell. While it is conceivable that some of these may become surface-exposed at some point, 33% of clusters seems high. The authors should perform validation experiments to check that they are not detecting intracellular proteins.

We would like to highlight a few points that we believe are important to consider when interpreting the GO enrichments in Fig. 2;

- Enrichments are relative to the total number of members for a GO term - they do not imply that proteins of a given GO term constitute a large group within a cluster. They can even account for a marginal number of proteins within a cluster. For example, if a hypothetical GO term has six members in the genome of the organism, and a hypothetical cluster contains 300 proteins including 3 of the hypothetical GO term, this would result in a highly significant enrichment despite accounting for only 1% of proteins within the cluster. In cluster 3 there is an enrichment of the term "Endoplasmic reticulum", cluster 3 proteins associated with this term make up ~20% of the cluster and ~3% of the total quantified proteins. In cluster 6 there is an enrichment of the term "lysosomal membrane", cluster 6 proteins associated with this term make up ~12% of the cluster and ~1% of the total quantified proteins.
- It is important to recognize that GO annotations are promiscuous and not mutually exclusive. Many proteins associated with the cell surface are also associated with terms related to intracellular membranes. For example, the cluster 6 protein ATP-binding cassette sub-family A member 2 (Abca2) has 75 associated GO terms, including endosome, lysosome membrane and plasma membrane.

- Imaging-based analysis of protein subcellular distribution (in the context of the human protein atlas) has established that half of all proteins localize to multiple compartments (Thul. et. al. 2017). Importantly the plasma membrane was identified as a highly dynamic structure that contains mainly multi-localizing proteins.
- GO terms constitute a useful resource and are frequently used, but they are also dynamic and incomplete; they aim to represent the current state of knowledge, but as of today this is of course far from perfect.

4. Synapses represent a small fraction of the neuronal surface, but the synaptic activity manipulations in Fig.5 (homeostatic plasticity) and Fig.6 (cLTP) will primarily affect the synaptic surface. Their surfaceome analysis may therefore underestimate local changes at the synapse.

We agree that this is an important point, the cell surface is a highly organized structure within micro- and nanodomains where proteins cluster to fulfill defined functions, most prominently at synapses. Upon synaptic plasticity there are certain effects that will primarily affect the synaptic surface, e.g. the area within the synaptic cleft, and others that will affect other regions on the neuronal surface. For example it has been shown that LTP leads to increased synaptic localization of AMPA receptors by surface diffusion into the synapse. At the same time there is evidence for exocytosis of AMPA receptors at domains outside of the synapse. Therefore, dissection of the nanoscale organization at synapses and on the neuronal surface will be crucial to understand neuronal functioning. We believe that the global analysis of changes occurring on the neuronal surface, as presented here, constitutes the first step towards that aim. The data presented here can serve as a resource to contrast further research targeting nanoscale organization of different domains on the neuronal surface.

5. The scaling experiment in Fig.5 and the cLTP experiment in Fig.6 lack validation to show that the manipulations were effective. Experimental validation should be provided. This is especially relevant for the scaling experiment, where the authors used 24hr treatment instead of the more commonly used 48hrs.

The aim of the scaling and cLTP experiments was to characterize surfaceome changes upon two widely used approaches to induce synaptic plasticity in culture, this should maximize the utility of our dataset for the scientific community.

Homeostatic scaling is commonly elicited by addition of TTX/BIC to neuron cultures for different time periods. From our perspective on the literature, we had the impression that the 24h treatment is often used;

- Dörrbaum et. al. 2020 (24h, 72h, 7 days)
- Schanzenbächer et. al. 2018 (2h, 24h)
- Schaukowitz et. al. 2017 (2h, 4h, 6h, 16h, 24h)
- Schanzenbächer et. al. 2016 (24h)
- Diering et. al. 2014 (24h, 48h)
- Ibata et. al. 2008 (1h, 2h, 3h, 4h, 24h)
- Wierenga et. al. 2005 (24h, 48h)

Importantly, Schanzenbächer et. al. (2016, Supplementary Fig. 1) already showed a significant change in miniature excitatory postsynaptic currents (mEPSCs) upon 24h of TTX and BIC treatment.

cLTP treatments of neuron cultures are also very commonly used, although very similar, we felt there is some variance in the composition of cLTP buffers used. Therefore we consulted with leading scientists in the field and followed their recommended protocol. Like most cLTP buffers, we use Glycine to stimulate the NMDA receptor, additionally Magnesium is removed from the media to inhibit the rapid Mg-block of the NMDA receptor. To increase efficiency of the treatment, we additionally slightly increase extracellular Calcium concentration and add BIC to counteract the unwanted side-effect caused by glycine binding to the inhibitory glycine receptor, block inhibition and increase spontaneous activity.

A hallmark of LTP, and cLTP treatment in culture, is the increased surface insertion of AMPA receptors. Therefore, we believe that the increased surface abundance of AMPA receptor subunits Gria1 and Gria2 measured by autoCSC are a clear indication that our cLTP treatment was effective.

Functionally, synaptic plasticity is best characterized using electrophysiological recordings. Such experiments require substantial expertise and are mostly performed by specialized research groups. Since we do not have the required resources to perform such experiments, we initiated a collaboration with the laboratory headed by Csaba Földy at the University of Zurich. We provide electrophysiological recordings after cLTP and homeostatic plasticity treatments and added new panels to Figures 5/6 and updated the main text with the following information:

- New Fig. 5 a,b,c: Electrophysiological recordings after homeostatic plasticity treatments
- New Fig. 6 b: Electrophysiological recordings after cLTP treatment
- “Increased miniature excitatory postsynaptic currents (mEPSCs) upon TTX and decreased mEPSCs upon BIC treatment were previously reported for 24h treatments (Schanzenbächer et al. 2016). In line with these results, we found increased fast EPSC amplitudes (**Fig. 5b**) as well as synaptic burst frequencies (**Fig. 5c**) comparing TTX with BIC treatment, indicative of differential synaptic scaling.”
- “To ascertain that our cLTP protocol leads to an increase in synaptic transmission, we performed electrophysiological recordings and confirmed increased fast EPSC amplitudes upon cLTP treatment (**Fig. 6a**).”

Figure 5: Homeostatic scaling (BIC/TTX)

Figure 6: cLTP

6. Experimental validation of some of the findings is required to determine the functional relevance of these findings. An interesting target protein the authors could focus on is Ptpn, which the authors show is regulated in both scaling and cLTP. Perturbation of Ptpn expression levels in combination with manipulation of activity can determine whether Ptpn is functionally required in scaling and cLTP .

We agree that Ptpn is a very promising protein to investigate its functional role in scaling and cLTP. However, we believe a further functional investigation is beyond the scope of this manuscript.

The aim is to provide a comprehensive resource of surfaceome changes during neuronal development and plasticity. Validation of a number of candidates with orthogonal approaches confirms our results. We tried to include Ptpn in these validation experiments, but we couldn't find a commercial antibody that is functional for live cell labeling.

Additionally we would like to highlight a few proteins that prominently come up in our experiments and were shown to be of functional relevance for synaptic plasticity;

- During homeostatic plasticity, we found a remarkable differential regulation of Nptx1, Nptx2 and Nptxr
 - Nptx1 was found as one of two proteins with coordinate regulation on surface and total abundance levels
 - Schaukowitch et. al. (2017) reported that inhibition of Nptx1 blocks synaptic upscaling
- Leaving aside the AMPA receptor, during cLTP treatment we found Wnt5a, PrnP and Adcy3 with increased surface abundance
 - Wnt5a KO mice show a functional LTP defect, Wnt5a was found essential for spatial learning and memory (Chen et al. 2017)
 - Aged PrnP KO mice show impaired LTP (Curtis et al. 2003)
 - Adcy3 KO mice show attenuated LTP and deficits in spatial navigation (Chen et al. 2016)

We believe a functional validation of one selected protein would not add to the overall utility of our resource for the community. Rather, we aim to make our data more accessible to enable researchers to use our data for further functional investigation. Therefore, we created an interactive webpage with cell surface abundance profiles during neuronal development for >1'000 proteins and total abundance profiles for >7'000 proteins readily available. Additionally, the web-based platform enables fast access to download surface and total proteotype data tables for neuronal development, homeostatic plasticity and cLTP via the following link:

- [Neurosurfaceome.ethz.ch](https://neurosurfaceome.ethz.ch)

Minor points:

1. How fast is extracellular N-glycosylation and does this bias the authors towards the identification of more stable CSPs?

Please see our response to Major point 2.

Modifications of glycan structures can be very fast, but biosynthesis of N-linked glycosylation is linked to protein biogenesis, and thereby likely linked to turnover of the respective protein. In addition, with autoCSC we take averaged snapshots of surface abundance over millions of cells, by measuring the abundance at a defined time point we are agnostic to stability or turnover of cell surface proteins.

2. The phrasing of the authors makes it seem like all synapse assembly factors and synaptic proteins are trafficked to the surface prior to synapse formation due to enrichment of these annotations in certain clusters. However, this does not mean that all of them behave this way and in fact this is apparent in Fig.3b and S8a.

We agree with Reviewer 1 and thank him for pointing this out, the phrasing in some sentences was misleading. We corrected the writing throughout in the main text.

3. The conclusion on page 9 that homeostatic scaling predominantly influences the postsynapse is at odds with the increase in surface SV2a/b and thus synaptic vesicle fusion (a presynaptic effect) the authors show. Please explain.

We thank the reviewer for pointing this out, we realized this wasn't clearly explained in the manuscript. TTX treatment blocks sodium channels and thereby inhibits firing of action potentials. From this immediate treatment effect we would expect a decrease in the amount of synaptic vesicle fusion events. Based on the previous finding

during neuronal development, where we observed an increase of SV2a/b on the surface upon increase of synapses and presumably synaptic activity, we hypothesize that changes in SV2a/b surface abundance can report on synaptic vesicle fusion events. Now, during homeostatic scaling, we observe a significant decrease in surface SV2a/b upon TTX treatment, which supports the notion that TTX treatment led to a decrease of synaptic vesicle fusion events. Blocking of neuronal activity by TTX however leads to synaptic upscaling - how are these two findings compatible? Previous reports showing that synaptic scaling predominantly affects the postsynapse could provide an explanation (Wierenga et al. 2005). If the synaptic scaling mechanism affects the synaptic vesicle release at the presynapse to a lower extent compared to the activity-blockage of TTX, then we would expect a decrease in synaptic vesicle fusion event, and we would attribute this finding to the activity blockage rather than synaptic scaling. This is compatible with the fact that TTX leads to synaptic upscaling, because postsynaptic scaling effects can still mediate the increase in synaptic strength.

The fact that we find this decrease in surface SV2a/b upon TTX treatment argues that the pharmacological TTX effect dominates over synaptic scaling mechanisms at the presynapse, leaving the postsynapse as predominant locus of scaling.

While Schanzenbächer et al. (2016) showed increased mEPSC amplitudes observed after 24h of TTX, we performed electrophysiological recordings in absence of TTX in the recording solution. This is an important distinction when comparing our electrophysiology data with autoCSC, because autoCSC reports on the situation with TTX present.

With the hope to make this more clear, we updated the main text as follows;

“Increased miniature excitatory postsynaptic currents (mEPSCs) upon TTX and decreased mEPSCs upon BIC treatment were previously reported for 24h treatments (Schanzenbächer et al. 2016). In line with these results, we found increased fast EPSC amplitudes (Fig. 5b) as well as synaptic burst frequencies (Fig. 5c) comparing TTX with BIC treatment, indicative of differential synaptic scaling.

As surface SV2a and SV2b levels increased with synapse formation and neuronal activity, we expected to observe a decrease in surface abundance upon TTX treatment. Indeed, using autoCSC, we found a significant reduction of both SV2a and SV2b compared with untreated controls (Fig. 5b,c), indicating a net decrease of neuronal synaptic vesicle release. In contrast, upon downscaling of GABAergic neurotransmission using BIC, which leads to onset of rapid neuronal firing, there was a significant increase in surface SV2a and SV2b (Fig. 5b,c), suggesting neuronal firing events lead to a net increase in synaptic vesicle fusion events. Importantly, autoCSC is expected to report on the status with TTX or BIC present, while our electrophysiological recordings were performed after the 24h treatment in absence of TTX or BIC. Considering the previous findings that homeostatic synaptic scaling predominantly influences the postsynapse (Wierenga et al. 2005), the observed changes in SV2a/b likely are a direct consequence of TTX or BIC treatment, rather than homeostatic scaling. ~~These results agree with previous findings that homeostatic synaptic scaling predominantly influences the postsynapse rather than presynaptic vesicle release and recycling⁶⁴.~~”

4. The authors claim that the surface/global proteome correlation is dependent on the time between measurements. Is this not influenced by the opposite change in correlation in the surface proteome between measurements as seen in fig 1C?

Figure 1c shows correlations between surface protein abundances for all time points. Differences in these correlations are based on changes in surface protein abundances.

Fig. 1c doesn't show a general change in correlation based on time between measurements, in fact, it shows that there are very strong changes between close time points during the early phase of development and fewer changes between distant time points at the later stages. For example, DIV4 and DIV8 are separated by 4 days and have a correlation coefficient of 0.87, while DIV8 and DIV20 are separated by 12 days and show the same correlation coefficient. Overall this shows that the changes in surface abundance are overall stronger during early development and the later time points are generally more similar.

Figure 4b,c and d should address a different question; are the changes observed on the surface abundance (and indicated in Fig 1c) correlated with the changes that we observe in the total abundance?

Therefore we plot the correlations of surface and total abundance changes either across the entire time series (a) or in relation to the time between measurements (b & c). The change in surface proteome between measurements strongly influences the surface/global correlation, as it is an integral part of the underlying data. The directionality of the surface changes doesn't matter for this analysis, we only ask whether they correlate with the total abundance changes.

5. Page 12: increased surface abundance does not necessarily equal increased exocytosis. Please rephrase.

We thank the reviewer for picking this up and corrected the following sentence;

"In contrast, we observed increased ~~exocytosis~~ surface abundance for another receptor-type tyrosine-protein phosphatase, Ptprc."

6. As a suggestion, synapse development in neuronal cultures is well established and the first paragraph of the results describing this could be condensed in order to start with novel findings that capture the attention of the reader.

We thank reviewer 1 for pointing out that synapse development in neuronal cultures is well established. Indeed, one reason we chose neuron cultures as a model system is that it is broadly used by the community (Pubmed search for "neuron culture" yields >110'000 entries). We agree with reviewer 1 that this part could be condensed. At the same time we received several questions regarding our neuron cultures in this review. Therefore, we decided to leave it as is for now.

We thank Reviewer 1 for his comments and suggestions that enabled us to provide an improved revised manuscript.

Reviewer #2 (Remarks to the Author):

Here van Oostrum et al., investigated the temporally-resolved surfaceome analysis of developing primary neuronal cultures. The study offers an interesting resource for the analysis of membrane proteins in primary neuronal cultures. However, the assays used are a combination of in-vitro and non-physiological conditions that makes difficult to extrapolate the results and conclusions, in particular to different forms of synaptic plasticity.

Indeed we used an in-vitro system, primary neuronal cultures, during development and in combination with frequently used pharmacological treatments to elicit synaptic plasticity. Compared to the in-vivo situation these are certainly non-physiological conditions and neurons in culture are different from neurons in a behaving animal. Nevertheless, many labs use surrogate methods for synaptic plasticity, such as cLTP, that reliably leads to increase in AMPA receptor exocytosis, synaptic trapping of receptors and long lasting increase in transmissions, which are all hallmarks of LTP. Even though cLTP is certainly different from LTP, it shows that these cultures have at least some apparatus to express plasticity and that it is useful to study the molecular mechanisms.

As a resource manuscript it fails to provide to the reader with a clear and ease to use dataset. I found extremely difficult to follow and evaluate the observations and conclusions of the manuscript since the tables provides only uniprot identifiers without indicating gene or protein id. While uniprot ids can be converted to any other identifier, the authors need to provide all the information in an appropriate format not just for the reviewers but for the readers.

We thank the reviewer 2 for this suggestion. We would like to make our manuscript as easy to use as possible, Therefore we added gene names to all tables and use them throughout the manuscript and in all figures. Additionally we created an interactive webpage with cell surface abundance profiles during neuronal development for >1'000 proteins and total abundance profiles for >7'000 proteins readily available and

searchable by gene name. It enables fast access to download surface and total proteotype data tables for neuronal development, homeostatic plasticity and cLTP via the following link:

- [Neurosurfaceome.ethz.ch](https://neurosurfaceome.ethz.ch)

While I cannot comment on the findings and conclusions of the manuscript without further evaluation of the molecules described, my main concern is on the utility of the dataset, in particular regarding different forms of synaptic plasticity. While the results can be of use within the context of neuronal cultures, the manuscript extrapolate the findings to synaptic plasticity. It is true that the glycine chem-LTP protocol have been extensively used in neuronal cultures, however these *in vitro* neurons do not show LTP and it is very well known that signaling mechanisms differ in cultured neurons and hippocampus slices. The authors claim: "Primary neuronal cultures have been used to study fundamental aspects of neuronal development, synapse formation, and synaptic plasticity" While this is true, it doesn't mean that this is the method of choice, or that our understanding of synaptic plasticity mechanisms are due to analysis in cell culture systems. Generally neuronal cultures are not the method of choice or most indicated to study synaptic plasticity. While insertion of AMPAR are necessary for LTP, measurement of this parameter of neuronal cultures is not a measure of LTP. Ideally, the assays should have been performed in slice preparations.

As outlined above, we generally agree that cLTP in cultures is different from LTP *in vivo*. We also agree that slice preparation would be more physiological and preferred. However, for a number of technical reasons we chose neuron cultures over slice preparations, one reason being that we are not convinced that uniform cell surface labeling can be achieved equally in a brain slice. We thank the reviewer for pointing out that cLTP have been used extensively in neuronal culture, we believe this is important to recognize despite the limitations. We would like to add that there are also advantages of neuronal cultures; their surfaceome is amenable for tagging, they are available in high cell numbers required for subcellular proteomics, their relative purity and approximate synchronizing of synapse formation during development *in vitro*.

For more than 20 years it is known that exocytosis is required for LTP (Lledo et al. 1998). An obvious and important question to ask is which proteins need to be exocytosed for LTP. While there have been many studies on the role of AMPA receptor exocytosis, it was never demonstrated that the AMPA receptor is one of, or even the critical cargo, and it still remains unclear which target(s) need to be exocytosed to produce LTP. Despite the long-standing importance of this question, until today, an unbiased and system-wide experiment showing changes in protein abundance on the cell surface upon LTP is lacking. While we cannot provide this experiment, our data using neuron cultures in combination with cLTP treatment is as close as technology permits for the time being. In our unbiased approach we recapitulate one hallmark of LTP, AMPA receptor exocytosis, and additionally we identify proteins with increased surface abundance that were shown to be functionally important for LTP. Albeit the differences, there are also shared molecular mechanisms between cLTP in neuron cultures and LTP in slices, it is not improbable to hypothesize that our findings are valid for LTP as well. Therefore, and despite the limitation of the model system, we believe our findings of 42 proteins with significantly different surface abundance upon cLTP treatment in cultures is valuable for the scientific community investigating molecular mechanisms of LTP, and can contribute to further research directed to solving this longstanding question.

A way forward might be to discuss the results within the context on an *in-vitro* system, which can produce chem-LTP in slices, without firmly stating that the results obtained underlie LTP.

We thank the reviewer for this suggestion and placed this important distinction prominently in the main text. We adjusted the wording throughout the manuscript in order to assure that it is clearly stated our data underlies cLTP treatment of neuron cultures and not LTP. Additionally we performed electrophysiological recordings after cLTP treatment in cultures to provide a more physiological correlate.

- *"While neuron cultures do not undergo LTP, they are frequently stimulated with cLTP treatments which reproduce hallmarks of LTP, such as AMPA receptor exocytosis, synaptic trapping of receptors and increase in synaptic transmission. To ascertain that our cLTP protocol leads to an increase in synaptic*

transmission, we performed electrophysiological recordings and confirmed increased fast EPSC amplitudes upon cLTP treatment (Fig. 6b)."

- *"Signalling mechanisms of synaptic plasticity differ in neuron cultures and hippocampal slices, therefore future studies are required to determine whether our findings from cLTP treatment of cultures are relevant for bona fide LTP. The data provided here from cLTP stimulation in vitro provides a number of avenues for further research."*
- New Fig. 6 a: Electrophysiological recordings after cLTP treatment
- *"To ascertain that our cLTP protocol leads to an increase in synaptic transmission, we performed electrophysiological recordings and confirmed increased fast EPSC amplitudes upon cLTP treatment (Fig. 6b)."*

Figure 6: cLTP

Another option is to show that the chem-LTP protocol can induce evoked excitatory postsynaptic currents (EPSCs) at synaptic connections between individual hippocampal neurons. This can be done using paired whole-cell recordings from pyramidal neurons. Then results can be discussed within this context, but not LTP.

We thank the reviewer for this suggestion. We preferred and followed the former.

A similar scenario is observed for homeostatic plasticity. Here, it is more complicated to address the results within the context of neuronal physiological processes. One, is the inherent problems to address the temporal scales of different forms of plasticity and second, the differences observed in culture and slice models.

While silencing neuronal activity with TTX has been extensively used to address processes involved with homeostatic plasticity, Hebbian and homeostatic forms of synaptic plasticity operate on vastly different time scales. Thus the field have been always struggled to explain how forms of homeostatic plasticity that develop over hours and days (here cultures are incubated by 24hs with TTX of Bic) can provide the negative feedback needed to regulate synaptic weights and stabilize activity in cells and circuits undergoing Hebbian synaptic plasticity, which can be induced in minutes or seconds (see chem LTP protocol used). The use of cell culture models makes more difficult to interpret within a physiological context. It is also known that homeostatic plasticity is differentially regulated under development. There is no rationale on the developmental stage selected or how this compares to a slice model. Both scenarios can be summarized in that TTX has no effect on mini frequency in cultured neurons (14 DIV or less), while the in-vivo infusion of TTX into the hippocampus of adult rodents does induce a profound increase in mini frequency while no effect on mini amplitude in CA1 pyramidal cells.

Irrespective of performing the assays in hippocampal slice or neuronal cultures. The protocols selected are of difficult comparison, by harshly blocking neuronal activity or mildly modulating GABAergic/Glutamate signaling. Again the physiological importance of blocking electrical activity for 24hs as a measure of homeostatic plasticity in a cell culture system is questionable. Again, it will be nice to have a physiological correlate to compare, more mechanistic, other than AMPAR insertion.

The aim of the scaling experiments was to characterize surfaceome changes upon two widely used approaches to induce synaptic plasticity in culture, this should maximize the utility of our dataset for the scientific community. Homeostatic scaling is commonly elicited by addition of TTX/BIC to neuron cultures, often used in mature DIV21+ cultures;

- Dörrbaum et. al. 2020
- Schanzenbächer et. al. 2018
- Schaukowitch et. al. 2017
- Schanzenbächer et. al. 2016
- Diering et. al. 2014
- Ibata et. al. 2008
- Wierenga et. al. 2005

We chose mature cultures because autoCSC averages across millions of cells, as such it is preferable if all cells/synapses are relatively homogenous. We reasoned that these experiments are best conducted when synapse formation has reached a plateau (compare with Supplementary Fig. 1) indicative of a more homogenous population of mature neurons and synapses, rather than during development where there might be more heterogeneity.

Schanzenbächer et. al. (2016, Supplementary Fig. 1) showed a significant change in miniature excitatory postsynaptic currents (mEPSCs) upon 24h of TTX and BIC treatment.

We provide electrophysiological recordings after homeostatic plasticity treatments and added new Panels to Figures 5/6 and updated the main text with the following information:

- New Fig. 5 a,b,c: Electrophysiological recordings after homeostatic plasticity treatments
- New Fig. 6 b: Electrophysiological recordings after cLTP treatment
- “Increased miniature excitatory postsynaptic currents (mEPSCs) upon TTX and decreased mEPSCs upon BIC treatment were previously reported for 24h treatments (Schanzenbächer et al. 2016). In line with these results, we found increased fast EPSC amplitudes (**Fig. 5b**) as well as synaptic burst frequencies (**Fig. 5c**) comparing TTX with BIC treatment, indicative of differential synaptic scaling.”

Figure 5: Homeostatic scaling (BIC/TTX)

Other:

The authors indicate that: on average 72% of synapses were excitatory. Does this mean that they determined a 28% of inhibitory synapses in their cultures? The number seems a bit high. How this number was calculated? What is the percentage of inhibitory neurons in the culture? Any bias toward somatostatin, pv+ neurons?

Numbers are highly variable between culture conditions. There is no report of inhibitory neurons in the manuscript. This is relevant in particular when considering the Bic protocol.

We determined an average of 72% of synapses to be excitatory and 28% to be inhibitory, as defined by colocalization of synapsin with either PSD95 or gephyrin. (See Supplementary Fig. 1)

We conducted additional experiments to characterize the composition of the culture with respect to inhibitory neurons. We added two panels to supplementary figure 1 and updated the main text with the following information:

- New supplementary Fig. 1 panels c and d
- “To further characterize our cortical culture system, we established the relative abundances of neurons and astrocytes. Of all cells that could be categorized based on Gfap and Map2 immunofluorescence, we found an average of 81% to be neurons (**Supplementary Fig. 1c-d**). Of those, we classified 23% on average as inhibitory neurons based on the presence of GAD1/2 mRNA at the cell soma (**Supplementary Fig. 1c-d**).”

There are a number of statements that are not clear or overstated, and the analysis do not show or help to have an in-depth understanding of the organization of surface proteins, for example:

“This analysis revealed several subnetworks. For example, the ephrin family is primarily associated with clusters 4 and 5 and cadherins are mostly associated with clusters 1 and 2. Cell adhesion molecules are relatively underrepresented in clusters 3 and 6, which involve proteins that increase in abundance starting at 12 DIV. This supports the notion that surface expression of cell adhesion molecules does not increase during synapse formation or synaptic activity”.

In general protein functions and families are overlooked and discussed within the context of trends in quantitative changes at different developmental stages. Together with the fact that only uniprot ids are provided, this makes the reading complicated and the conclusions difficult to evaluate.

For example, the authors say:

mRNAs that encode cluster 4 and 5 proteins are enriched in astrocytes however, they said before:

“For example, the ephrin family is primarily associated with clusters 4 and 5” and they conclude: This supports the notion that surface expression of cell adhesion molecules does not increase during synapse formation or synaptic activity.

The, the authors are now discussing ephrins (which are expressed also in neurons) together with a cluster enriched in astrocytes. This might be ok from the point of view of clustering molecules with a particular criteria, but the conclusions not only are very confusing for the general reader, but also can be very misleading depending on the considerations for clustering analysis.

We thank the reviewer for this comment, we revised and condensed this section and removed the statements regarding cell adhesion molecules.

Is not clear if the clustering analysis consider just trends in protein levels or also cell types. The cultures have at least three main cell types, excitatory, inhibitory neurons and astrocytes. A relative quantitation for each cell type will be nice.

We conducted experiments to characterize the composition of the culture, we added two panels to supplementary figure 1 and updated the main text with the following information:

- New supplementary Fig. 1 panels c and d (see above)
- “To further characterize our cortical culture system, we established the relative abundances of neurons and astrocytes. Of all cells that could be categorized based on Gfap and Map2 immunofluorescence, we found an average of 81% to be neurons (**Supplementary Fig. 1c-d**). Of those, we classified 23% on average as inhibitory neurons based on the presence of GAD1/2 mRNA at the cell soma (**Supplementary Fig. 1c-d**).”

The authors said: The total abundance is determined by the interplay of protein synthesis and degradation, and regulated surface trafficking likely accounts for the higher extent of differential expression on the surface. It is not clear if the authors considered or normalized their results against the number of synapses present.

We did not normalize against the number of synapses present. The data of our experiment comparing the surfaceome during development from DIV2 to DIV20 is normalized on peptide level prior to N-glycopeptide capture and also bioinformatically by median equalization within the statistical framework of MSstats. As such our data provides relative quantification of each surface protein across different timepoints normalized to total protein abundance.

The sentence quoted above is independent of the number of synapses present, it states that the protein abundance is a function of protein synthesis and protein degradation, while the surface abundance contains a third regulatory element, regulated surface trafficking. The aim was to emphasize that this regulation can account for the observation that the surfaceome shows more changes and higher magnitude of protein fold-change differences compared to the global proteome.

Suggesting neuronal firing events lead to a net increase in synaptic vesicle fusion events. These results agree with previous findings that homeostatic synaptic scaling predominantly influences the postsynapse rather than presynaptic vesicle release and recycling. It is not clear why synaptic vesicle fusion events inform on postsynapse rather than presynaptic vesicle release

We thank the reviewer for pointing this out, we realized this wasn't clearly explained in the manuscript. TTX treatment blocks sodium channels and thereby inhibits firing of action potentials. From this immediate treatment effect we would expect a decrease in the amount of synaptic vesicle fusion events. Based on the previous finding during neuronal development, where we observed an increase of SV2a/b on the surface upon increase of

synapses and presumably synaptic activity, we hypothesize that changes in SV2a/b surface abundance can report on synaptic vesicle fusion events. Now, during homeostatic scaling, we observe a significant decrease in surface SV2a/b upon TTX treatment, which supports the notion that TTX treatment led to a decrease of synaptic vesicle fusion events. Blocking of neuronal activity by TTX however leads to synaptic upscaling - how are these two findings compatible? Previous reports showing that synaptic scaling predominantly affects the postsynapse could provide an explanation (Wierenga et al. 2005). If the synaptic scaling mechanism affects the synaptic vesicle release at the presynapse to a lower extent compared to the activity-blockage of TTX, then we would expect a decrease in synaptic vesicle fusion event, and we would attribute this finding to the activity blockage rather than synaptic scaling. This is compatible with the fact that TTX leads to synaptic upscaling, because postsynaptic scaling effects can still mediate the increase in synaptic strength.

The fact that we find this decrease in surface SV2a/b upon TTX treatment argues that the pharmacological TTX effect dominates over synaptic scaling mechanisms at the presynapse, leaving the postsynapse as predominant locus of scaling.

While Schanzenbächer et al. (2016) showed increased mEPSC amplitudes observed after 24h of TTX, we performed electrophysiological recordings in absence of TTX in the recording solution. This is an important distinction when comparing our electrophysiology data with autoCSC, because autoCSC reports on the situation with TTX present.

With the hope to make this more clear, we updated the main text as follows;

“Increased miniature excitatory postsynaptic currents (mEPSCs) upon TTX and decreased mEPSCs upon BIC treatment were previously reported for 24h treatments (Schanzenbächer et al. 2016). In line with these results, we found increased fast EPSC amplitudes (Fig. 5b) as well as synaptic burst frequencies (Fig. 5c) comparing TTX with BIC treatment, indicative of differential synaptic scaling.

As surface SV2a and SV2b levels increased with synapse formation and neuronal activity, we expected to observe a decrease in surface abundance upon TTX treatment. Indeed, using autoCSC, we found a significant reduction of both SV2a and SV2b compared with untreated controls (Fig. 5b,c), indicating a net decrease of neuronal synaptic vesicle release. In contrast, upon downscaling of GABAergic neurotransmission using BIC, which leads to onset of rapid neuronal firing, there was a significant increase in surface SV2a and SV2b (Fig. 5b,c), suggesting neuronal firing events lead to a net increase in synaptic vesicle fusion events. Importantly, autoCSC is expected to report on the status with TTX or BIC present, while our electrophysiological recordings were performed after the 24h treatment in absence of TTX or BIC. Considering the previous findings that homeostatic synaptic scaling predominantly influences the postsynapse (Wierenga et al. 2005), the observed changes in SV2a/b likely are a direct consequence of TTX or BIC treatment, rather than homeostatic scaling. ~~These results agree with previous findings that homeostatic synaptic scaling predominantly influences the postsynapse rather than presynaptic vesicle release and recycling⁵⁴.~~”

It is not clear what the authors means by:

Furthermore, these results indicate that there are relatively few qualitative alterations to the surfaceome during maturation in culture, suggesting that regulation rather affects the quantitative surface abundance.

Most proteins (>75%) we identified at all timepoints, 85% at all times points beyond the first two. This indicates that there are relatively few changes in the composition of the surfaceome, and we suggest that rather the quantitative abundance of these proteins changes during development. For clarification we added the relevant reference (Fig. 1b) to the sentence.

we found that Gria1 and Gria2 had significantly increased surface abundance upon stimulation (Fig. 6b, c) indicating that the cLTP protocol triggers AMPA receptor exocytosis.

This is extensively reported in multiple systems including chem LTP in neuronal cultures

We thank the reviewer for highlighting this fact.

The strongest responder was the receptor type tyrosine-protein phosphatase Ptpn.

Here it is not clear if Ptprn is discussed within the context of postsynaptic or presynaptic membrane. This is important because of pre/postsynaptic site in synaptic plasticity. PTPR proteins have been usually described at the presynaptic site. It is not clear if the authors are claiming a role for Ptprn at the postsynaptic site in relation to AMPAR trafficking. If so, it will be good to show that Ptprn is present and have a role at the postsynaptic site or if the observed correlation is to the presynaptic site. In fact this might occur with a number of the reported proteins, without having a knowledge of their localization. I acknowledge that this is not easy to perform, but also the statements should be more careful and in line with the assays performed.

The reviewer raises an important point, which side of the synapse is the protein located? Our analysis strategy is agnostic to this information and we do not make any claims about the subsynaptic localization. We revised the two sentences where we comment on subsynaptic localization of proteins identified in the cLTP treatment experiment to make this point more clear:

“We identified receptors with known localizations from both pre- and post-synaptic sites (e.g., adenosine receptor A1 and cannabinoid receptor 1) and of various neurotransmitter types. The majority of receptors identified were previously described as postsynaptic G-protein coupled receptors (...)”

The neuronal surfaceome interaction network.

There are no major details on how the interaction network was built. Is this a protein-protein interaction (PPI) network? How the information was curated. PPI networks usually include all sorts of information, obtained by different methods (Y2H, IP, overexpression, etc.), developmental stages, different species, cell types, etc... Is extremely important to perform a manual curation of the data used. In particular for membrane proteins where IP assays usually include “pieces” of membrane with all sorts of non-interacting membrane proteins.

The interaction network figures are not clear I don't know what the edges of the network means, or which ones are the nodes connected. This info might be within the uniprot ids but it is not clear. For example, does the figure implies a PPI between Grm1, Gria1? Where this info is coming from? Or between Unc5a/d-Dcc/Dscam? Nrnx2-Nrxn3 (these are just some examples) there are many edges that are unknown PPI.

The *in silico* interaction network is created by mapping the quantified surface proteins to a network using interaction data from the STRING database, as outlined in the main text. The STRING interactions are derived from five main sources, genomic context predictions, high-throughput lab experiments, (conserved) co-expression, automated text mining and previous knowledge in databases. Therefore, we cannot expect every edge to represent a physical protein-protein interaction, rather a loosely defined functional link. For this reasons, the STRING database is very comprehensive, and we find it useful for this purpose because it effectively visualizes subnetworks of protein families with related functions. In the figure legend, we additionally specify that the edges represent a STRING confidence score of 0.7, which is considered high confidence by STRING. Briefly, this score indicates the confidence of an interaction (rather than strength or specificity), and is computed by integration of different types of evidence. This is described in detail here (von Mering et al. 2005). We added this reference to the figure legend.

The decrease of Nlgn1 seems strange as it has been reported to increase through development in excitatory synapses. Are there any other Nlgn family members increasing through neuronal development? Or any comments on this?. Again it is difficult to follow the data with just a figure

The illustration of cluster profiles in Fig. 3a represents the cluster averages, for simplicity. Importantly, as can be seen in Fig. 2a, there is some variance in how close each protein agrees with the cluster average. As a threshold to assign a protein to a cluster, it needs to pass a membership cutoff value (set at 0.5) which indicates how well the protein profile agrees with the cluster average. This assures that only proteins are associated with the cluster they are close to, based on their profile. In order to make the profiles of individual proteins comparable, they need to be normalized, and this is done using their overall variance. Nlgn1 illustrates one caveat with this procedure, proteins with very small overall variability (i.e. there is no significant change during the time series) may show a similar profile as proteins with overall higher variability (but the same trend) and are clustered together. While we remove proteins with no or very low variability from the dataset, this is a somewhat artificial threshold and some

proteins with little variability will be included as well. For Nlgn1 this is the case, it shows no significant change throughout the time series, with a slight downward trend.

To enable the reader fast access to the profiles of each protein, we created the new webpage neurosurfaceome.ethz.ch where the development profiles of individual proteins are now easily accessible by their gene name.

Methods:

What's the reasoning behind using bicuculine in a Mg free solution with 200uM Gly?

Compare also the answer above to Reviewer 1:

cLTP treatments of neuron cultures are also very commonly used, although very similar, we felt there is some variance in the composition of cLTP buffers used. Therefore we consulted with leading scientists in the field and followed their recommended protocol. Like most cLTP buffers, we use Glycine to stimulate the NMDA receptor, additionally Magnesium is removed from the media to inhibit the rapid Mg-block of the NMDA receptor. To increase efficiency of the treatment, we additionally slightly increase extracellular Calcium concentration and add BIC to counteract the unwanted side-effect caused by glycine binding to the inhibitory glycine receptor, block inhibition and increase spontaneous activity.

What's the rationale behind performing c- LTP assays at DIV 14 and not on DIV 20?

With autoCSC we average across millions of cells and even more synapses. Due to the high basal level of surface AMPA, the expected increase of AMPA receptor surface abundance upon cLTP is not very high, for Gria1 and Gria2 we find an approx. 35% increase. Therefore we aimed at maximizing the "potential" for potentiation by cLTP stimulation. At DIV14 we would expect a higher fraction of immature synapses that can eventually be potentiated more than mature synapses at DIV20.

We thank Reviewer 2 for his comments and suggestions that enabled us to provide an improved revised manuscript.

Reviewer #3 (Remarks to the Author):

The MS by Oostrum et al. reports the application of a previously published proteomic method for quantifying cell surface proteins to neuronal cell cultures. In particular, the authors investigate changes in the cell surface composition during neuronal development (in culture), and synaptic plasticity experiments. While many of the observed changes correlate with overall changes in protein abundance, they identify notable exceptions, ie proteins whose cell surface abundance appears to be strongly regulated by trafficking. The MS thus reports on a generally useful and scalable experimental approach to study the neuronal 'surfaceome', and provides good sample applications providing new insights into neuronal surface dynamics.

The MS is well written, the experiments are technically sound, and the results are well presented. The approach is timely and powerful. Nevertheless, I would like to raise two important issues that currently limit the impact of the study.

We sincerely thank reviewer 3 for acknowledging our experiments and manuscript.

1. The MS repeatedly states that the method captures only 'bona fide' cell surface proteins, and no intracellular proteins. However, as discussed further below, a substantial proportion of intracellular proteins are also recovered, many of them from the endoplasmic reticulum and from lysosomes. Using the provided data and external subcellular localisation annotation, I estimate the 'purity' at around 80% plasma membrane proteins –

good, but by no means pure. Hence, any claims to 100% specificity need to be toned down considerably, and all findings in the study need to be re-evaluated in light of the reduced specificity, where necessary.

We do not achieve nor do we wish to claim 100% specificity. Even more, we believe it is not possible to achieve 100% specificity for any enrichment method coupled to mass spectrometry (MS) for different reasons, among others the probabilistic nature of identification, usually provided at 1% FDR. Furthermore with increased sensitivity of MS instrumentation, we generally identify more unspecific background. If one would optimize for specificity, the simplest solution would be to use a less sensitive MS, leading to identification of only/predominantly the most abundant and highly enriched proteins.

This illustrates the importance of the unique feature of CSC among cell surface protein enrichment methods, the ability to distinguish the peptides of interest (e.g. the enriched peptides) from the background noise, using a modification of the identified peptides - similar to a phosphoenrichment. Because we identify de-glycosylated peptides (by the mass shift resulting from PNGase F cleavage at an asparagine located within a NXS/T motif) initially tagged on live cells by a cell-impermeable linker, we can conclude that this peptide was exposed to the extracellular space. Therefore we conclude that this protein features at least one extracellular amino acid and would thus qualify as *bona fide* cell surface protein. It is important to recognize this because in contrast, other cell surface protein enrichment strategies cannot distinguish whether the identified peptides were extracellularly exposed or originate from background noise and require prior knowledge (e.g. cell surface protein lists generated by other means) to filter their identifications.

The phrasing that our method captures *bona fide* cell surface proteins thus should highlight the ability of CSC to provide experimental evidence for cell surface localization without requiring *a priori* knowledge.

The phrasing should not however, claim that we reach 100% specificity.

In order to avoid such confusion, we removed this phrasing from the main text and adjusted the wording where we suspected it could be misleading. Additionally we added a statement to the discussion that specifies that autoCSC is not 100% specific but produces false positives as well. See also our response to the further comments regarding specificity below.

2. The study uses cultured cortical neurons, which are in fact a mixture of astrocytes and neurons. This very important point is mentioned rather late in the results section, but has major implications for the entire study. Without orthogonal information, it is not possible to decide which of the identified proteins are neuronal and which ones are from astrocytes. But this is critical for any inferences regarding the neuronal surface. No attempt was made to quantify the relative proportions of astrocytes and neurons. Importantly, the proportions might change during the course of the differentiation, and this may well account for some of the observed changes. The authors need to address this point in detail. The short foray into gene expression (Supp. Figure 8) is in my opinion insufficient.

Since the proposed approach has the potential to become a widely used 'gold-standard' for investigating neuronal surface dynamics, it will be very important to discuss and address these limitations.

We thank the reviewer for the positive comment on our experiments.

As mentioned above, we conducted additional experiments to characterize the cortical cultures in detail; The cortical cultures used here are primary cultures and as such they may contain any cell type present in the tissue of origin, which is predominantly neurons but also astrocytes. Ratios between cell types do not change during development because neurons are postmitotic and proliferation of glia is inhibited by addition of cytosine arabinoside (AraC) on (see methods). Because the treatment is at DIV4, there might be a bias in the first time point (DIV2). In order to highlight this caveat, we added this information to the main text. Additionally we conducted experiments to characterize the composition of the culture, we added two panels to supplementary figure 1 and updated the main text with the following information:

- New supplementary Fig. 1 panels c and d
- “To further characterize our cortical culture system, we established the relative abundances of neurons and astrocytes. Of all cells that could be categorized based on Gfap and Map2 immunofluorescence, we found an average of 81% to be neurons (**Supplementary Fig. 1c-d**). Of those, we classified 23% on average as inhibitory neurons based on the presence of GAD1/2 mRNA at the cell soma (**Supplementary Fig. 1c-d**).”
- “In order to preclude glial overgrowth, the cultures were treated with cytosine arabinoside (AraC) on DIV4.”

Specific comments

The authors sub-cluster the 943 identified ‘surface’ proteins into six groups, based on common changes during the time course (Fig. 2). They claim that all of these proteins are ‘bona fide’ cell surface proteins.

We do not claim 100% surface specificity (see above) and removed and revised the wording.

I have used Uniprot subcellular localisation annotation for rat proteins to annotate the six clusters for a rough compositional analysis. Uniprot provided localisation annotations for 381 of the 943 proteins. Of these 30 were annotated as ER, 25 as Golgi, 17 as lysosomal, 4 as nuclear, 1 as mitochondrial – so in sum $77/381 = 20\%$ with a clear non-plasma membrane localisation. So even assuming that all other annotated proteins are indeed plasma membrane localised, one would estimate the proportion of PM proteins at around 80%.

This analysis is also in line with the estimates of PM purity by Weekes et al. (2010, J Biomol Tech.21:108-115), who compared different cell surface labelling and capture approaches, including the present one (59-85% estimated purity).

Please note that I do not wish to argue the exact level of purity here; the important point is that there is a substantial pool of intracellular proteins, and this needs to be considered.

Nevertheless we are not convinced that the presented analysis is suitable to accurately estimate the proportion of cell surface proteins. The reasons are similar to those mentioned above with respect to GO terms;

- Uniprot annotations are neither perfect nor the gold-standard of protein subcellular localization data; this is also apparent by the fact that only 40% of proteins had an annotation at all.
- As GO terms, they are not mutually exclusive. Many proteins associated with the cell surface are also associated with terms related to intracellular membranes. For example, Slc3a2 is associated with 5

different localizations; lysosome membrane, cell membrane, apical cell membrane, cell junction and melanosome.

- Imaging-based analysis of protein subcellular distribution (cell atlas in the context of the human protein atlas) has established that half of all proteins localize to multiple compartments (Thul. et. al. 2017). Importantly the plasma membrane was identified as a highly dynamic structure that contains mainly multi-localizing proteins.

Weekes et. al. compare a number of cell surface labeling approaches, this includes data from three experiments using a modified version of CSC. All three experiments use protein-level enrichment, while CSC enriches glycopeptides, therefore this is not CSC. The authors also do not call it CSC in the paper. Additionally, two of the three experiments are based on on-bead tryptic digestion, making it even more different. Only one experiment uses PNGase F release (but after protein-level enrichment), and it is not clear whether the identified peptides were filtered for presence of deamidated asparagines within the NXS/T glycosylation motif. Furthermore, the experiment using PNGase F identified a total of 21 proteins. We acknowledge that the study is 10 years old, but the percentage of plasma membrane protein annotations of these 21 proteins (achieved by a method which is not CSC) is not helpful as comparison. In summary, the results from Weekes et. al. are not comparable to the autoCSC workflow presented here.

The proportion of plasma membrane proteins varies widely between clusters. Clusters 1, 2, 4, and 5 are highly enriched in plasma membrane proteins, but 3 and 6 contain only few. Indeed, the authors remark on this, and try to explain it:

"Clusters 3 and 6 share a curious feature: Both show an acute increase in abundance starting at 12 DIV (Fig. 2a), the time point when synapse formation increases considerably (Supplementary Fig. 1b). Interestingly, both clusters are enriched in proteins associated with intracellular membranes, but the two clusters are enriched in membranes from different organelles (cluster 3: "endoplasmic reticulum part", "membrane enclosed lumen", cluster 6: "vacuolar membrane", "lysosomal membrane", "coated vesicle"). These intracellular membranes potentially appear at the surface at increased rates due to alterations in surface trafficking. Associated annotations point to functions related to degradation or clearing of biomolecules from the surface (cluster 3: "proteolysis in cellular catabolic process", "ERAD Pathway", cluster 6: "catabolic process", "autophagy", "hydrolase activity") but also synthesis of new biomolecules (cluster 3: "glycoprotein biosynthetic process", "macromolecules biosynthetic process") (Fig. 2b). These findings could reflect acute changes in the dynamic interplay of biosynthesis, endosomal degradation, recycling, and surface trafficking, all important for control of synapse formation and modulation of synaptic plasticity."

While it is not impossible for an integral membrane protein of the ER to appear at the cell surface transiently, major steady state pools of ER proteins at the cell surface are, to my knowledge, not a common occurrence (as the data presented here would suggest); how a luminal ER protein would be retained at the cell surface would also require an explanation.

We cannot extrapolate from the presented enrichments to proportions in the clusters. Enrichments are relative to the total number of members for a GO term - they do not imply that proteins of a given GO term constitute a large group within a cluster. They can even account for a marginal number of proteins within a cluster. For example, if a hypothetical GO term has six members in the genome of the organism, and a hypothetical cluster contains 300 proteins including 3 of the hypothetical GO term, this would result in a highly significant enrichment despite accounting for only 1% of proteins within the cluster. In cluster 3 there is an enrichment of the term "Endoplasmic reticulum", cluster 3 proteins associated with this term make up ~20% of the cluster and ~3% of the total quantified proteins. In cluster 6 there is an enrichment of the term "lysosomal membrane", cluster 6 proteins associated with this term make up ~12% of the cluster and ~1% of the total quantified proteins. Therefore, the data doesn't suggest major steady state pools of ER proteins at the surface - however multi localization of ER and cell surface seems frequent, see below (Thul et. al. 2017).

It is important to recognize that GO annotations are promiscuous and not mutually exclusive. Many proteins associated with the cell surface are also associated with terms related to intracellular membranes. For example,

the cluster 6 protein ATP-binding cassette sub-family A member 2 (Abca2) has 75 associated GO terms, including endosome, lysosome membrane and plasma membrane.

GO terms constitute a useful resource and are frequently used, but they are also dynamic and incomplete; they aim to represent the current state of knowledge, but as of today this is of course far from perfect.

As mentioned above, Imaging-based analysis of protein subcellular distribution (in the context of the human protein atlas) has established that half of all proteins localize to multiple compartments (Thul et. al. 2017). Importantly the plasma membrane was identified as a highly dynamic structure that contains mainly multi-localizing proteins.

And what would be the function of trafficking biosynthetic ER proteins or a nuclear pore complex member (NUP210) to the cell surface? A much simpler explanation is either a limited unwanted permeation of the cell membrane by the labelling reagent, resulting in labelling of intracellular proteins; or perhaps the labelling of cell debris/dead cell remnants in the cell culture medium.

So for a proper interpretation of the data in the context of neuronal development, I think it will be necessary to acknowledge the labelling of some non-surface proteins, and curate the list of detected proteins accordingly.

Solely for the reason that we cannot assign a cell surface specific function we cannot discard our experimental observation. The reviewer mentions Nup210, should we exclude this protein because obviously a nuclear pore complex member belongs to the nuclear pore and not the cell surface?

The database COMPARTMENTS predicts for NUP210 besides nucleus and endoplasmic reticulum (confidence score 4) also localization to the plasma membrane (confidence score 3). NUP210 also does not contain an ER retention signal. While the cell atlas doesn't include NUP210 it lists 135 proteins with multi localization to the nucleus and the plasma membrane. And these 135 do not include the insulin receptor, which only recently was shown to translocate to the nucleus to interact with DNA (Hancock et al. 2019).

However, we do not wish to argue about the subcellular localization of NUP210, but we would like to illustrate that it is not impossible that NUP21 does appear on the cell surface, and in absence of the ground truth, we should report all CSC identified peptides as experimental observations.

Furthermore, many databases and annotations are curated from different sources, predominantly cancer cell lines and rarely these include primary neurons. As a resource this makes them very useful, but not necessarily as a ground-truth background to curate data generated from primary neuron cultures. Neurons are an extreme case among cells with many specializations that deviate from principles found elsewhere, for example neuronal dendrites largely lack canonical Golgi membrane. In this light, the most unexpected findings might even be the most interesting, as they could provide insights into neuronal specialization.

Therefore we are not convinced to curate our identifications based on database annotations. There is currently no gold-standard resource for neuronal cell surface proteins that could provide the grounds for a curation.

More important than providing absolute specificity in protein identifications, is the ability to provide meaningful quantitative information by discriminating cell surface localized proteins to the same protein localized intracellularly. In order to detect subtle changes in surface abundance it is required that the contamination by intracellularly localized proteins is minimal. For example, it has been estimated that 60%-70% of the AMPA receptors are present intracellularly, tagging of even a subset of the intracellular AMPA receptors would render detection of a <50% increase in surface AMPA hardly possible, especially considering the technical variance associated with any enrichment workflow.

We can detect multiple proteins with changing cell surface abundance upon cLTP treatment, during which no changes were detected in total protein abundance during the 20min time window. The ability to uncover quantitative localization specific differences - in absence of global changes - is much more important than reaching absolute specificity in identifications, and illustrates the high surface specificity.

As mentioned above, we removed this phrasing from the main text and adjusted the wording where we suspected it could be misleading. Additionally we added a statement to the discussion that specifies that autoCSC is not 100% specific but also produces false positives.

Related to this problem:

P6/7

"Proteins from clusters 3 and 6 are encoded by mRNAs more highly expressed in neurons than in astrocytes, whereas mRNAs that encode cluster 4 and 5 proteins are enriched in astrocytes."

In light of my comments above, I find this rather alarming – the most neuron-specific proteins are in the clusters that contain the smallest proportion of surface proteins. So their behaviour will not inform on cell surface processes.

In response to previous comments we removed this sentence and revised the section to make it more clear.

In any case, the statement that the most neuron specific clusters contain the smallest proportion of surface proteins is not accurate.

- The most neuron enriched proteins are not in cluster 4 and 5,
 - They are in cluster 1, together with cluster 2 they make up almost half of the neuron enriched proteins
 - Cluster 4 and 5 are relatively more populated compared with astrocyte enriched proteins
- We cannot make the statement that cluster 3 and 6 contain the smallest proportion of surface proteins based on the GO enrichments of Fig.2
- Also, the focus is not on the most neuron specific in general (i.e. the highest reads of all transcripts) but the highest fold-change difference to astrocytic transcripts

P5

How does Supp. Figure 6 show that 6 is the optimum number of clusters? To me it looks like the minimum of the graph is at cluster size of 8 or 9? Please clarify.

The desired cluster number is not at the minimum of the graph. The optimal cluster number is after the “drop” in minimum centroid distance followed by a slower decrease in minimum centroid distance for higher cluster numbers. This is explained in more detail here (Schwämmle and Jensen 2010).

P8 and Figure 4C

"Together, this demonstrates that for many proteins the total abundance pool correlates with the surface pool over the entire time course of neuronal differentiation in culture but there are considerable fluctuations over time. These findings suggest that changes in surface trafficking influence surface abundance generally on short time scales, whereas total protein abundance modulates the surfaceome over long time periods."

A simpler explanation is that cultures with close time points will be more similar (as Figure 1C shows), and hence there will be fewer genuine differences, and likely of smaller magnitude. These will therefore also not correlate so well. But if you look at the most distant time points, the differences will be largest, and hence easier to quantify accurately, resulting in better correlation.

As mentioned above, for some comparisons close time points are very similar - but not always. The data shows that there are very strong changes between close time points during the early phase of development and fewer changes between distant time points at the later stages. For example, DIV4 and DIV8 are separated by 4 days and have a correlation coefficient of 0.87, while DIV8 and DIV20 are separated by 12 days and show the same

correlation coefficient. Overall this shows that the changes in surface abundance are overall stronger during early development and the later time points are generally more similar.

However, the reviewer raises an interesting point. Our measurements are confounded by technical variance. For very small real differences this will have a greater influence compared to relatively larger differences. When comparing surface and total protein pools, we would expect that technical variance does not correlate (assuming it is random) and this could result in low correlation in absence of real differences.

In order to demonstrate that technical variance does not invalidate our findings, we included two figures showing the same observation, Fig. 4c and Fig. 4d. Fig 4c shows correlation coefficients of Surface and Total log₂FC differences for all proteins - and will be influenced to some degree by the technical variance that over proportionally affects small differences. In Fig. 4d however, we only look at a subset of proteins, those that reached the significance cutoff, set at fold-change of 1.5 and adj. p-value of 0.05. Then, we asked what is the percentage of significant surface differences that are also observed on the total abundance level. Therefore, this analysis should only include genuine differences, and because we compare percentages, it is also agnostic to the absolute number of differences. As shown in Fig. 4d, these percentages show the same distribution as the correlations in Fig. 4c.

To test this, the authors could for example compare the median magnitude of 'significant' differences between adjacent time points – is this smaller than the median difference for the most distant time points?

The magnitude of significant differences is higher for distant time points, but this increase is equal for both surface (green) and total (blue) abundance (see plot below). This reflects the fact that the majority of surface proteins follow the same global trend throughout the time series, either increasing or decreasing (Fig. 2). The concurrent increase of both surface and total analysis indicates that both analyses are capable of detecting significant changes of smaller magnitudes - and these differences are genuine, because they are above the FC and p value threshold - but the overlap is smaller on shorter versus longer time intervals.

It is also interesting to compare with the experiments using pharmacological plasticity induction, because they extend the interval time in the lower range, 24h for homeostatic scaling and 20min for cLTP.

TTX and BIC treatment are relatively drastic, and after only 24h of synaptic scaling we find many changes of relatively high magnitude (compare Supplementary Fig. 14). Still, the overlap between Surface and Total abundance is merely 2%, even lower than the 13% found for adjacent time points (48h) during development.

For cLTP the time interval is so narrow that it would be very difficult for a cell to achieve significant changes of the total protein level. Accordingly, we find no overlap of changes between surface and total protein pools.

Table S2

I have tried and failed to extract the 128 proteins that show significant differences between surface and proteome pool changes from Table S2. Please provide these in a separate list, with gene names annotated, to make this a more useful resource.

We thank the reviewer for pointing this out. These can be found in Table S2 sheet 10. We highlighted their location in sheet 1 summary tab and additionally highlighted them with color in sheet 10. We added gene names to all supplementary tables.

Fig 6i

Please show a control of a protein not changing to make this figure more convincing. See also my next comment.

We performed additional validation experiments for two proteins that showed no significant difference after cLTP stimulation by autoCSC.

- New Supplementary Fig. 16b-e: Validation experiments for cLTP
- “Additionally we could confirm for two subunits of the GABA_A receptor (GABRA1 and GABRA2) that they do not change significantly upon cLTP (**Supplementary Fig. 17b-e**).”

Fig 6i, legend

Two different statistical tests were used for different proteins in the same experiment – this is highly unusual and will need to be justified.

T-test was used when the data showed a normal distribution. This was not the case for GluA, therefore Mann-Whitney test was more appropriate.

Discussion:

"Here, we present an initial blueprint of how many, and which, proteins are present on the surface of neurons. First, neurons come in different populations and our rat cortical neuronal culture system contains a mixture of these cells. Furthermore, our cultures contain predominantly neurons, but also astrocytes. We addressed this question partially by use of transcriptomic data; a more complete delineation could be achieved by combination with cell-type specific metabolic labeling approaches."

See my major point 2 above. Since no attempts were made to quantify the neuronal and astrocyte populations, any conclusions regarding cell surface dynamics in neurons are limited.

See answer above regarding characterization of the cortical culture.

We thank Reviewer 3 for his comments and suggestions that enabled us to provide an improved revised manuscript.

REVIEWER COMMENTS

Reviewer #1 (Remarks to the Author):

I appreciate the efforts the authors have made to address my comments. They have satisfactorily addressed my minor points. Regarding my major points, the added data in response to points #1 and 2 have improved the manuscript. The added data for point #5 however is difficult to interpret and raises doubts on the efficacy of the pharmacological treatments used for surfaceome analysis in plasticity conditions. This should be addressed.

Major Point 1. The added data on neuron/glia composition and excitatory/inhibitory neuron ratio (supplementary Figure 1) have clarified this point and improved the manuscript. Please indicate in the figure legend at what age in culture these quantifications were performed.

Major Point 2. The added validation experiments on surface abundance changes after induction of homeostatic plasticity have improved the manuscript.

Major Point 3. The point on the presence of intracellular proteins was also raised by another reviewer. I agree with the authors that GO terms are problematic. However, I do think it is important to perform additional validation experiments to determine to what extent CSC detects non-surface proteins.

Major Point 4. OK.

Major Point 5. The experimental validation of homeostatic plasticity treatments and cLTP raises questions on the efficacy of these treatments. The observed effects of TTX and BIC on EPSC amplitude are very small and none of these are significantly different from the control condition, in contrast to what is observed by others. Although the authors refer to a number of studies that have used similar treatments (e.g. Schanzenbacher et al Neuron 2016), the data shown in Figure 5 is difficult to compare with these studies, as it appears that the authors recorded spontaneous EPSCs in the absence of TTX, whereas the studies they refer to recorded mEPSCs in the presence of TTX. As the authors indicate to their response to minor point #3, the actual CSC experiment was performed in the presence of TTX. Thus, it is not clear how the electrophysiological data shown here demonstrates efficacy of the homeostatic plasticity treatment used or how it is relevant as for the surfaceome analysis. These experiments should be repeated in the presence of TTX so mEPSC

amplitudes can be compared for control, TTX and BIC conditions to judge efficacy and proper induction of homeostatic scaling.

The scalebar in Figure panel 5A indicates that the EPSCs in the TTX condition are extremely large, is this correct? What does this bursting upon TTX treatment signify? I could not find similar observations in other studies using 24h homeostatic scaling treatments.

The effect of cLTP on EPSC amplitude appears very small and mainly due to a small number of observations with higher amplitudes. What is the p-value here?

Major Point 6. OK.

Reviewer #2 (Remarks to the Author):

I'm happy with most of the answers to my comments. In my view the manuscript is more clear and greatly improved

Reviewer #3 (Remarks to the Author):

The authors have done a lot of work to address the reviewers' suggestions, including my own, and the manuscript is much improved.

However, in one point I still disagree with the authors' view. This concerns the specificity of surface labelling. The authors argue that their method exclusively captures proteins that are exposed at the cell surface, as they focus on N-glycosylated peptides:

Authors' statement:

'Because we identify de-glycosylated peptides (by the mass shift resulting from PNGase F cleavage at an asparagine located within a NXS/T motif) initially tagged on live cells by a cell-impermeable linker, we can conclude that this peptide was exposed to the extracellular space.'

This is where I disagree. N-glycosylation occurs inside the cell; hence, glycosylated proteins are also found in the ER, Golgi, endosomes, and lysosomes. Therefore, the authors' argument only holds as long as the chemical linker is 100% membrane impermeable. To my knowledge this is practically impossible to achieve. Even if the chemical itself is very impermeant, cells can be a bit leaky/damaged,

and also actively endocytose extracellular material (with residual activity even at low temperatures). Furthermore, cell debris from dead cells is present in abundance in the culture supernatants from cultured cells, and even rigorous washing will not eliminate this problem. As a result, it is highly likely that the labelling will include a proportion of intracellular proteins, stemming from either of the above sources. I don't think the authors can completely rule out that this happens. In my opinion, this does not invalidate the overall excellent method; but I think it is misleading not to acknowledge the possibility of low-level non-specific capture.

I propose to at least mention this caveat – so I would like the author to add a statement to their paragraph on page 6:

[...] Interestingly, both clusters are enriched in proteins associated with intracellular membranes, but the two clusters are enriched in membranes from different organelles (cluster 3: “endoplasmic reticulum part”, “membrane enclosed lumen”, cluster 6: “vacuolar membrane”, “lysosomal membrane”, “coated vesicle”). These intracellular membranes potentially appear at the surface at increased rates due to alterations in surface trafficking. Associated annotations point to functions related to degradation or clearing of biomolecules from the surface (cluster 3: “proteolysis in cellular catabolic process”, “ERAD Pathway”, cluster 6: “catabolic process”, “autophagy”, “hydrolase activity”) but also synthesis of new biomolecules (cluster 3: “glycoprotein biosynthetic process”, “macromolecules biosynthetic process”) (Fig. 2b) . These findings could reflect acute changes in the dynamic interplay of biosynthesis, endosomal degradation, recycling, and surface trafficking, all important for control of synapse formation and modulation of synaptic plasticity 11,39–41.

How about adding:

‘Alternatively, some of these proteins may have been identified through labelling of small amounts of intracellular proteins from cell debris present in the culture supernatant, or very low levels of cell permeabilization by the linker.’

Provided the authors include this or a similar statement, I can recommend publication of the manuscript.

Point-by-point Response NCOMMS-19-34283B

Reviewer #1 (Remarks to the Author):

I appreciate the efforts the authors have made to address my comments. They have satisfactorily addressed my minor points. Regarding my major points, the added data in response to points #1 and 2 have improved the manuscript. The added data for point #5 however is difficult to interpret and raises doubts on the efficacy of the pharmacological treatments used for surfaceome analysis in plasticity conditions. This should be addressed.

Major Point 1. The added data on neuron/glia composition and excitatory/inhibitory neuron ratio (supplementary Figure 1) have clarified this point and improved the manuscript. Please indicate in the figure legend at what age in culture these quantifications were performed.

We thank Reviewer 1 for acknowledging our efforts in detailed characterization of the primary neuron cultures using immunocytochemistry and RNA-FISH to determine the relative abundance of excitatory neurons, inhibitory neurons and astrocytes. We added the age of the cultures to the figure legend.

Major Point 2. The added validation experiments on surface abundance changes after induction of homeostatic plasticity have improved the manuscript.

We thank Reviewer 1 for acknowledging our orthogonal experimental validation of the CSC data for homeostatic plasticity and neuronal development using microscopy.

Major Point 3. The point on the presence of intracellular proteins was also raised by another reviewer. I agree with the authors that GO terms are problematic. However, I do think it is important to perform additional validation experiments to determine to what extent CSC detects non-surface proteins.

CSC is described in more detail in the previous original and follow-up publications (Wollscheid et. al. Nature Biotechnology, 2009; van Oostrum & Müller et al, Nature Communications 2019). In addition to the sentence on specificity already added to the discussion, we added a second sentence as requested by Reviewer 3 (see below).

Major Point 4. OK.

Major Point 5. The experimental validation of homeostatic plasticity treatments and cLTP raises questions on the efficacy of these treatments. The observed effects of TTX and BIC on EPSC amplitude are very small and none of these are significantly different from the control condition, in contrast to what is observed by others.

The p-values are 0.0007 (BIC-vs-TTX), 0.043 (Control-vs-TTX) and 0.086 (Control-vs-BIC). There was a mistake in the figure legend, the supplementary data file contains the correct information.

We corrected the figure legend accordingly.

Although the authors refer to a number of studies that have used similar treatments (e.g. Schanzenbacher et al Neuron 2016), the data shown in Figure 5 is difficult to compare with these studies, as it appears that the authors recorded spontaneous EPSCs in the absence of TTX, whereas the studies they refer to recorded mEPSCs in the presence of TTX. As the authors indicate to their response to minor point #3, the actual CSC experiment was performed in the presence of TTX. Thus, it is not clear how the electrophysiological data shown here demonstrates efficacy of the homeostatic plasticity treatment used or how it is relevant as for the surfaceome analysis.

These experiments should be repeated in the presence of TTX so mEPSC amplitudes can be compared for control, TTX and BIC conditions to judge efficacy and proper induction of homeostatic scaling.

We thank Reviewer 1 for his/her comments on our experiments. For the following reasons we are not convinced that the addition of mEPSC recordings for TTX and BIC treatments would significantly improve our manuscript;

- Schanzenbächer et. al. (Neuron 2016) used the exact same treatment as we use in this paper (24h BIC or TTX) and already showed a significant change in mEPSCs.
 - An exact replication of an experiment goes against the legally binding 3R principles of animal research and would require substantial justification in order to be approved by the veterinary office of the canton Zürich.
 - We and others intentionally apply widely used perturbations that emerged as standards in the field. This ensures comparability and enables compliance with 3R principles. Asking for repetitions of validation experiments for every publication defies this principle.
- The CSC data on homeostatic plasticity recapitulates previous findings, demonstrating that the treatment was effective.
 - In agreement with our data, it was previously reported that both the surface abundance and total abundance of Nptx1 increases after 24 hours of TTX treatment (Immunocytochemistry for surface abundance, and Western blot for total abundance, Schaukowitz et. al. 2017, also found by Dörrbaum et. al. 2020)
 - Diering et al. found no significant changes for Gria1, Gria2, or Gria3 surface abundance after 24 hours of TTX treatment, in agreement with our findings. (surface biotinylation followed by Western Blot, Diering et. al. 2014).
- Our experimental validation of the CSC data using live-cell labeling and microscopy further demonstrates that the treatment was effective.
 - Supplementary Figure 15.
- Addition of these experiments would not affect the main findings of this paper, but take up a considerable amount of time and resources for our collaborator. Especially considering the current pandemic with labs being completely or partially closed and may open only slowly with restrictions, we don't see sufficient justification considering the added value for the manuscript.
- We are not convinced by the rationale provided by Reviewer 1.

- Reviewer 1 states that because “the actual CSC experiment was performed in the presence of TTX”, he suggests that “these experiments should be repeated in the presence of TTX so mEPSC amplitudes can be compared for control, TTX and BIC conditions”.
- Recording of mEPSCs requires the presence of TTX, irrespective of the prior experimental treatment.
- The CSC experiments involving homeostatic plasticity were not performed in the presence of TTX. For the condition involving TTX, the CSC experiments report on the status with chemical treatment present because the labeling is performed on ice, preserving the state of the proteotype.
- However, for BIC and control conditions TTX is never present during the CSC experiments. Therefore, we are not convinced by this argumentation.

The scalebar in Figure panel 5A indicates that the EPSCs in the TTX condition are extremely large, is this correct? What does this bursting upon TTX treatment signify? I could not find similar observations in other studies using 24h homeostatic scaling treatments.

The effect of cLTP on EPSC amplitude appears very small and mainly due to a small number of observations with higher amplitudes. What is the p-value here?

It indicates scaling in cell excitability. The p-value for cLTP-vs-Control is 0.025.

Major Point 6. OK.

Reviewer #2 (Remarks to the Author):

I'm happy with most of the answers to my comments. In my view the manuscript is more clear and greatly improved

We thank Reviewer 2 for acknowledging our efforts and new data.

Reviewer #3 (Remarks to the Author):

The authors have done a lot of work to address the reviewers' suggestions, including my own, and the manuscript is much improved.

We thank Reviewer 3 for acknowledging our efforts and new data.

However, in one point I still disagree with the authors' view. This concerns the specificity of surface labelling. The authors argue that their method exclusively captures proteins that are exposed at the cell surface, as they focus on N-glycosylated peptides:

Authors' statement:

'Because we identify de-glycosylated peptides (by the mass shift resulting from PNGase F cleavage at an asparagine located within a NXS/T motif) initially tagged on live cells by a cell-impermeable linker, we can conclude that this peptide was exposed to the extracellular space.'

This is where I disagree. N-glycosylation occurs inside the cell; hence, glycosylated proteins are also found in the ER, Golgi, endosomes, and lysosomes. Therefore, the authors' argument only holds as long as the chemical linker is 100% membrane impermeable. To my knowledge this is practically impossible to achieve. Even if the chemical itself is very impermeant, cells can be a bit leaky/damaged, and also actively endocytose extracellular material (with residual activity even at low temperatures). Furthermore, cell debris from dead cells is present in abundance in the culture supernatants from cultured cells, and even rigorous washing will not eliminate this problem. As a result, it is highly likely that the labelling will include a proportion of intracellular proteins, stemming from either of the above sources. I don't think the authors can completely rule out that this happens. In my opinion, this does not invalidate the overall excellent method; but I think it is misleading not to acknowledge the possibility of low-level non-specific capture.

We acknowledge the possibility of low-level non-specific capture. In addition to the statement already added to the discussion, we added the sentence proposed by Reviewer 3 (see below).

I propose to at least mention this caveat – so I would like the author to add a statement to their paragraph on page 6:

[...] Interestingly, both clusters are enriched in proteins associated with intracellular membranes, but the two clusters are enriched in membranes from different organelles (cluster 3: "endoplasmic reticulum part", "membrane enclosed lumen", cluster 6: "vacuolar membrane", "lysosomal membrane", "coated vesicle"). These intracellular membranes potentially appear at the surface at increased rates due to alterations in surface trafficking. Associated annotations point to functions related to degradation or clearing of biomolecules from the surface (cluster 3: "proteolysis in cellular catabolic process", "ERAD Pathway", cluster 6: "catabolic process", "autophagy", "hydrolase activity") but also synthesis of new biomolecules (cluster 3: "glycoprotein biosynthetic process", "macromolecules biosynthetic process") (Fig. 2b). These findings could reflect acute changes in the dynamic interplay of biosynthesis, endosomal degradation, recycling, and surface trafficking, all important for control of synapse formation and modulation of synaptic plasticity 11,39–41.

How about adding:

'Alternatively, some of these proteins may have been identified through labelling of small amounts of intracellular proteins from cell debris present in the culture supernatant, or very low levels of cell permeabilization by the linker.'

Provided the authors include this or a similar statement, I can recommend publication of the manuscript.

We added the statement and thank Reviewer 3 for his/her valuable comments.